# Genome-scale CRISPR screen identifies TMEM41B as a multi-function host factor required for coronavirus replication

**Limeng Sun[1,2]☯, Changzhi Zhao[3]☯, Zhen Fu[1,2], Yanan Fu[1,2], Zhelin Su[1,2], Yangyang Li[4], Yuan Zhou[3], Yubei Tan[1,2], Jingjin Li[3], Yixin Xiang[1,2], Xiongwei Nie[3], Jinfu Zhang[3], Fei Liu[4], Shuhong Zhao[3,5], Shengsong Xie®[3]\*, Guiqing Peng®[1,2]\***

**1** State Key Laboratory of Agricultural Microbiology, College of Veterinary Medicine, Huazhong Agricultural University, Wuhan, P. R. China, **2** Key Laboratory of Preventive Veterinary Medicine in Hubei Province, The Cooperative Innovation Center for Sustainable Pig Production, Wuhan, P. R. China, **3** Key Laboratory of Agricultural Animal Genetics, Breeding and Reproduction of Ministry of Education & Key Lab of Swine Genetics and Breeding of Ministry of Agriculture and Rural Affairs, Huazhong Agricultural University, Wuhan, P. R. China, **4** Joint International Research Laboratory of Animal Health and Food Safety & Single Molecule Nanometry Laboratory (Sinmolab), Nanjing Agricultural University, Nanjing, P. R. China, **5** Hubei Hongshan Laboratory, Frontiers Science Center for Animal Breeding and Sustainable Production, Wuhan, P. R. China

☯ These authors contributed equally to this work.
\* ssxie@mail.hzau.edu.cn (SX); penggq@mail.hzau.edu.cn (GP)

**Data Availability Statement:** Raw sequencing data and processed counts data for sgRNA libraries that support the findings of this study have been deposited in National Center for Biotechnology

## Abstract

Emerging coronaviruses (CoVs) pose a severe threat to human and animal health world-wide. To identify host factors required for CoV infection, we used α-CoV transmissible gastroenteritis virus (TGEV) as a model for genome-scale CRISPR knockout (KO) screening. Transmembrane protein 41B (TMEM41B) was found to be a *bona fide* host factor involved in infection by CoV and three additional virus families. We found that TMEM41B is critical for the internalization and early-stage replication of TGEV. Notably, our results also showed that cells lacking TMEM41B are unable to form the double-membrane vesicles necessary for TGEV replication, indicating that TMEM41B contributes to the formation of CoV replication organelles. Lastly, our data from a mouse infection model showed that the KO of this factor can strongly inhibit viral infection and delay the progression of a CoV disease. Our study revealed that targeting TMEM41B is a highly promising approach for the development of broad-spectrum anti-viral therapeutics.

## Author summary

Coronaviruses (CoVs) pose a severe threat to human and animal health. Identification of host genes essential for transmissible gastroenteritis virus (TGEV) infection may reveal novel therapeutic targets and strengthen our understanding of CoV disease pathogenesis. We performed genome-wide and sub-pooled CRISPR screens in porcine kidney cells with TGEV. We identified numerous candidate host factors for TGEV infection. Considering its extremely strong effect on TGEV infection, we focused our efforts on characterizing the roles that TMEM41B play in the virus replication cycle. Our results revealed that

Information's Gene Expression Omnibus and are accessible through GEO Series accession number: https://www.ncbi.nlm.nih.gov/geo/query/acc.cgi?acc=GSE169113.

**Funding:** This work was supported by the National Natural Science Foundation of China (grants No.: 31873020, to GQP), China National Funds of Distinguished Young Scientists (grants No.: 32125037, to GQP), and National Natural Science Foundation of China (grants No.: 32072685, to SSX). The funders had no role in study design, data collection and analysis, decision to publish, or preparation of the manuscript.

**Competing interests:** The authors have declared that no competing interests exist.

TMEM41B contributed to CoV internalization and the formation of replication organelles. TMEM41B is critical for the replication of TGEV and other RNA viruses. Our mouse infection model provides strong evidence that knocking out the TMEM41B gene can inhibit viral infection and delay the progression of a CoV disease. Our study identified potential therapeutic targets for common RNA viruses and reveals host factors that regulate susceptibility to highly pathogenic CoVs.

## Introduction

As enveloped positive-strand RNA viruses, coronaviruses (CoVs) have genetically evolved into the following four major genera: α, β, γ, and δ. CoVs cause various diseases in mammals and birds, ranging from enteritis in pigs to potentially lethal human respiratory infections [1,2]. For instance, severe acute respiratory syndrome coronavirus 2 (SARS-CoV-2), which belongs to the β-CoV genera, is a causative viral pathogen that causes upper respiratory diseases, fever, and severe pneumonia in humans [3] and currently presents a global public health threat. Transmissible gastroenteritis virus (TGEV), together with several human CoVs (*e.g.*, HCoV-NL63 and HCoV-229E), belongs to the α-CoV genera. TGEV causes severe diarrhea, vomiting, and dehydration, with mortality rates of 100% in piglets less than 2 weeks old, which causes significant economic losses in the pork industry worldwide [4,5]. Mouse hepatitis virus (MHV), belonging to the β-CoV genera, is a natural pathogen that specifically infects mice. MHV can cause numerous diseases through different inoculation routes, such as the liver, gastrointestinal tract, and central nervous system [6,7]. The MHV-A59 strain can produce moderate to severe hepatitis, mild to moderate acute meningoencephalitis, and chronic demyelination in C57BL/6 weanling mice [8]. MHV infection of mice is considered one of the suitable animal models for the study of CoV-host interactions [9–12].

Although numerous CoVs exist, they employ a relatively conservative replication strategy. The positive single-stranded RNA genome needs to be released into the cytosol to initiate infection. To do so, CoVs bind to host-encoded receptors [13–16] through their spike (S) glycoproteins, and the S2 domain adopts a post-fusion conformation after being cleaved by the host's various proteases for membrane fusion [17–20]. Subsequently, the viral RNA is released into the cytoplasm where the viral RNA is translated to generate the viral replication/transcription complex [21,22], and virions are then assembled and bud. Various host protein-coding genes are involved as critical factors in the CoV replication process [23]. Therefore, the identification of essential genes for the interaction between the virus and host cells can facilitate the development of targeted drugs for treating diseases related to CoV infections [24–26].

CRISPR screening has proven to be a powerful tool for identifying host factors necessary for infection by different viruses, such as SARS-CoV-2 [27,28], influenza A virus (IAV) [29], and Japanese encephalitis virus (JEV) [30]. In the present study, genome-wide CRISPR-based knockout (KO) screening was performed to specifically identify host factors required for TGEV replication in porcine kidney-15 (PK-15) cells. Among other identified host factors, we focused on TMEM41B and showed that this protein is required for the infection by CoV and other viruses. Our results revealed that the KO of TMEM41B inhibits the internalization and early-stage replication of CoV. Notably, we found that TMEM41B contributes to the formation of CoV replication organelles (ROs). Our study emphasized the significance of TMEM41B for CoV infection *in vivo* and *in vitro* which indicates that TMEM41B may be a vulnerable target for the development of anti-CoV therapeutics.

## Results

### A genome-scale CRISPR screen identified host factors associated with CoV TGEV infection

To identify host factors involved in α-CoV TGEV replication, we generated a porcine genome-scale CRISPR/Cas9 knockout (PigGeCKO) cell collection (in the PK-15 background) as described in a previous study [30] and performed a genome-wide loss-of-function genetic screen. Prior to screening, we first examined TGEV-induced cell death following infection at multiplicity of infection (MOI) values of 0, 0.001, 0.01, and 0.1. We observed cytopathic effects for the different MOIs at approximately 3 days after TGEV infection (S1 Fig). Here, we chose the optimal titer for TGEV-induced cell death in PK-15 cells with an MOI of 0.001. Subsequently, we performed screening to identify host factors that modulate susceptibility to TGEV-induced cell death. The overall PigGeCKO screening strategy is illustrated in Fig 1A. We first conducted three rounds of TGEV challenge, employing untreated Cas9 stably expressed (PK-15-Cas9) cells as a negative control to confirm cell death caused by TGEV infection in each round. At an MOI of 0.001, all untreated TGEV-infected PK-15-Cas9 cells died, whereas a small number of viable cells from the TGEV-infected PigGeCKO cell collection were observed. The surviving mutant cells were collected and used for subsequent TGEV challenge rounds. Lastly, small guide RNA (sgRNA) constructs in surviving mutant cells from the three rounds were polymerase chain reaction (PCR) amplified and deep sequenced.

Our screening revealed that a total of 335 unique sgRNA sequences, which targeted 317 unique protein-coding genes, were present in at least ~0.1% of the total number of cells analyzed from each round of TGEV challenge (S1 Data). As expected, aminopeptidase N (ANPEP), the already known functional receptor of TGEV [31,32], was significantly enriched in all three rounds of CRISPR screening (Fig 1B–1D). Among the top 0.1% of sgRNAs, multiple sgRNAs targeting the same gene were identified. For example, in the second challenge rounds, two and three sgRNAs hit the *ANPEP* and *TMEM41B* genes, respectively (S1 Data). Note that in general, a high number of sgRNA reads typically indicates the KO of the targeted gene, which confers strong resistance to TGEV-induced cell death but does not affect cell growth. A comparison of enriched sgRNAs from the positive selection CRISPR screening revealed that 76.7% of the very highly enriched (*i.e.*, ≧ 0.1%) sequences were common to the first, second, and third challenge rounds, and that 82.8% of the highly enriched (≧ 0.5%) sequences were common to all three rounds (Fig 1E and 1F). These results suggested that the candidate host factors related to TGEV infection are reliable based on our CRISPR screening strategy.

### TMEM41B is a host factor required for CoV TGEV replication

We also performed a focused-CRISPR library screening which examined the 79 top-ranked candidate host factors from the PigGeCKO screen; note that we excluded any sgRNA targeting the previously identified *ANPEP* receptor (S2 Data). In addition, a high virus titer (MOI = 1) was used to identify KO cells with a strong ability to inhibit TGEV infection. The 10 top-ranking candidates after the third and fifth challenge rounds were *FUT8*, *TMEM41B*, *DYRK1A*, *MLST8*, *etc.* from highest to lowest (Fig 2A). In addition, four sgRNAs were found to hit *TMEM41B* in the third and fifth challenge rounds, indicating that hitting the target should be reliable (Fig 2A). Afterward, five candidates (*i.e.*, *ANPEP*, *TMEM41B*, *LPP*, *TAX1BP*, and *BARHL2*) were selected for further analysis. Sanger sequencing confirmed that each of these single-clone-originated KO cell lines had one or more nucleotide deletions predicted to cause a frameshift mutation in the coding regions of the targeted gene (a non-integer multiple of 3)

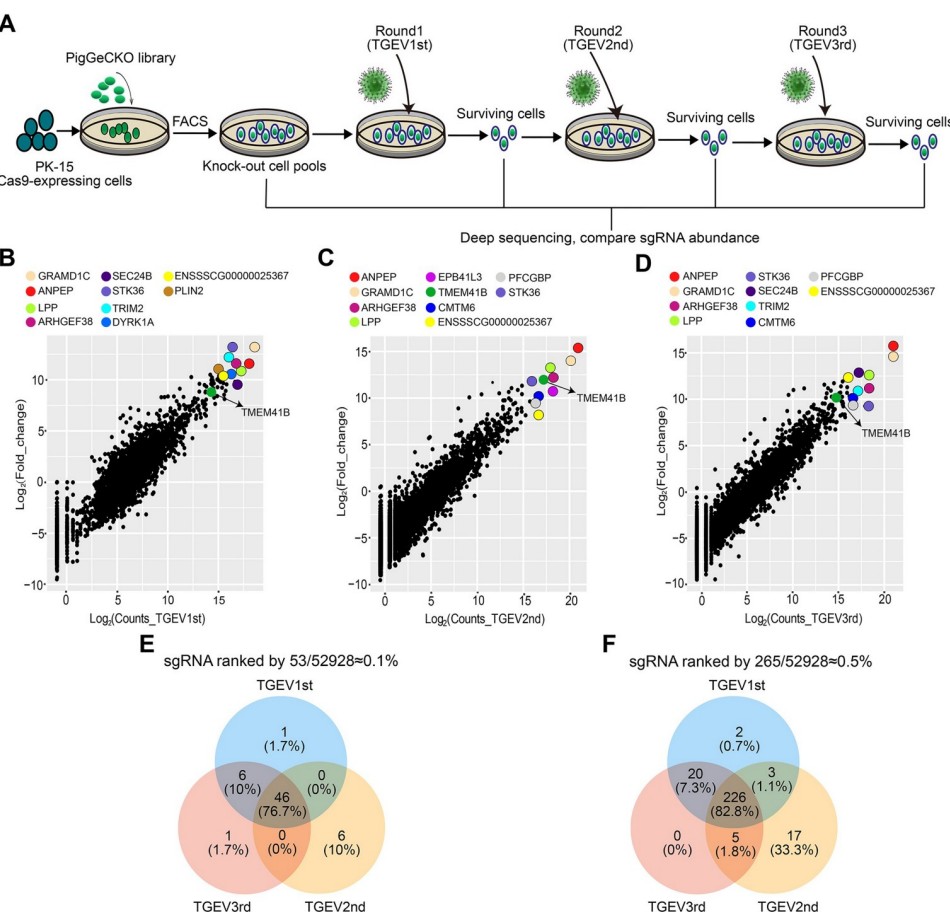

**Fig 1. Genome-wide CRISPR screening to identify genes associated with alpha-CoVs TGEV-induced cell death.**
**(A)** Identification of TGEV host factors using the porcine genome-scale CRISPR/Cas9 knockout (PigGeCKO) library. Transformed PK-15-Cas9 cells were either mock-treated or challenged with TGEV (MOI = 0.001). Surviving cells from each round of virus challenge were isolated, with the subsequent PCR amplification and sequencing of sgRNA. **(B-D)** Scatter plots showing sgRNA-targeted sequence frequencies and the extent of enrichment in transformed PK-15-Cas9 cells (mock-treated versus TGEV infected) in three rounds of TGEV screening; **(B)** first, **(C)** second and **(D)** third, respectively. Counts_TGEV1st/Counts_TGEV2nd and Counts_TGEV3rd are the average reads from the paired-end sequencing for each round. Log2(fold change) is the median log 2 ratio between normalized sgRNA count of TGEV challenged and mock-treated populations. **(E and F)** Venn diagrams indicate the scope of overlapping enrichment in specific sgRNA targeting sequences for the three TGEV screening rounds amongst the top **(E)** ~0.1% and **(F)** ~0.5% of averaged sgRNAs reads. FACS, Fluorescence Activated Cell Sorting; TGEV, Transmissible gastroenteritis virus; sgRNA, small guide RNA.

(S2 Fig). We found that the individual KO of these genes reduced viral loads in PK-15 cells (Fig 2B). This finding coincides with the immunofluorescence assays showing that the expression of the TGEV-encoded nucleocapsid (N) protein in five KO cell lines was reduced or undetectable following TGEV infection at an MOI of 0.1 (Fig 2C).

We prioritized *TMEM41B* for further mechanistic studies on TGEV replication, because the KO of this gene conferred a relatively strong ability to inhibit TGEV (MOI = 1) replication. The ability of TMEM41B KO cells to inhibit TGEV replication was near the level that we observed for ANPEP KO cells (Fig 2B and 2C). Western blot showed that the KO of TMEM41B can block the accumulation of the TGEV N protein (Fig 2D). Subsequently, plaque assays revealed that TMEM41B KO cells can remain viable after TGEV infection with MOIs of 0.001, 0.01, 0.1, 1, and 10 (Fig 2E). Moreover, as shown in Fig 2F, knocking out *TMEM41B*

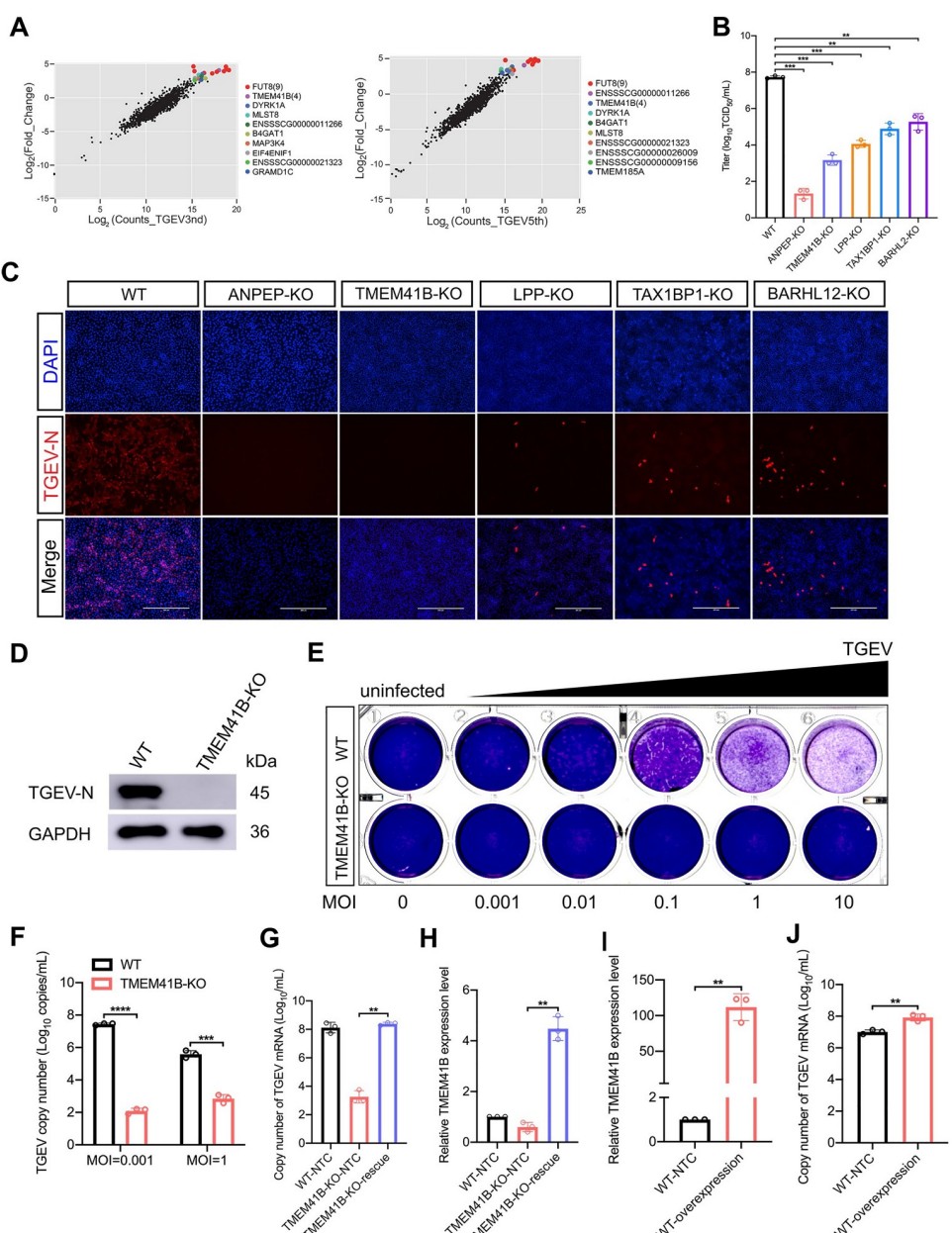

**Fig 2. TMEM41B is a host factor required for TGEV replication. (A)** Verification of candidate genes enriched in the genome-wide CRISPR screen using a second, focused-CRISPR library screen. Scatter plots compare sgRNA targeted sequence frequencies and the extent of enrichment in the transformed PK-15-Cas9 cells (mock-treated versus TGEV infected) for the third (TGEV3rd) and fifth (TGEV5th) rounds of TGEV screening. Counts_TGEV3rd and Counts_TGEV5th represent the average values of the read counts from paired-end sequencing. Log2(fold change) is the median log 2 ratio between normalized sgRNA count of TGEV challenged and mock-treated cell populations. **(B)** Quantification of virus infectivity (TCID$_{50}$) in culture supernatant collected 24 hpi from TGEV-infected (MOI = 0.1) WT and KO (*ANPEP*, *TMEM41B*, *LPP*, *TAX1BP1*, *BARHL12*) cell lines. **(C)** Immunofluorescence assays for detection of the TGEV N protein in WT cells and five selected genes (*ANPEP*, *TMEM41B*, *LPP*, *TAX1BP1*, *BARHL12*) KO cells following infection with TGEV (MOI = 0.1) at 24 hpi. Scale bar, 200 μm. **(D)** Western blot assay to detect the TGEV N protein expressed in *TMEM41B* KO and WT cells following infection with TGEV (MOI = 1) at 24 hpi. GAPDH used as an internal control gene. **(E)** Viral quantification by plaque assays. *TMEM41B* KO and WT cells were seeded into 24-well culture plates and infected with TGEV at different MOIs (0, 0.001, 0.01, 0.1, 1 and 10). Plates stained with 1% crystal violet to view plaques. **(F)** *TMEM41B* KO and WT cells were infected with TGEV at different MOIs (0.001 and 1), TGEV N copy number was assessed by absolute quantitative real-time PCR. **(G and H)** Rescue assays for WT, TMEM41B-KO and TMEM41B-KO-rescue cells infected with TGEV (MOI = 1). RT-qPCR assay for determination of **(G)** relative mRNA level of TMEM41B and **(H)** absolute mRNA level of TGEV N gene. TMEM41B-KO-rescue:

reconstituted TMEM41B in TMEM41B KO cells. **(I and J)** Overexpression of TMEM41B in PK-15 control cells following infection with TGEV (MOI = 1). RT-qPCR assay for the determination of **(I)** relative mRNA level of TMEM41B and **(J)** absolute mRNA level of the TGEV N gene. PK-15-NTC: Transfection of pcDNA3.1 empty vector in WT cells; PK-15-overexpression: Transfection of pcDNA3.1-TMEM41B vector in WT cells. WT, wild-type; KO, knockout; MOI, multiplicity of infection; kDa, kilodaltons; DAPI, 4',6-diamidino-2-phenylindole. $**P < 0.01$; $***P < 0.001$; $****P < 0.0001$. $P$ values were determined by two-tailed Student's $t$-tests. Data are represented as a percentage mean titer of triplicate samples relative to WT control cells ± S.D.

significantly reduced TGEV RNA replication at MOIs of 0.001 and 1. To detect whether TMEM41B is required for TGEV infection, we constructed a CRISPR-resistant TMEM41B-KO-rescue cell line, which stably expressed the *TMEM41B* gene on its KO cells (S2 Fig). As shown in Fig 2G and 2H, the TGEV replication on TMEM41B-KO-rescue cells can be restored completely as compared with the wild-type (WT) cells. Furthermore, overexpression of *TMEM41B* in WT cells can promote TGEV replication (Fig 2I and 2J). These results indicated that TMEM41B as a critical gene participates in TGEV replication. In addition, no differences were detected in the rates of DNA synthesis or cell proliferation between TMEM41B KO and WT cells in EdU fluorescence assays or cell proliferation assays (MTS), respectively (S3B and S3C Fig).

Subsequently, the role in TGEV replication of other TMEM41B homologous proteins with a VTT domain (collectively referred to as DedA domains), such as Transmembrane Protein 41A (TMEM41A) and vacuole membrane protein 1 (VMP1), was determined [33]. Compared with TMEM41B KO cells, we found that knocking out TMEM41A did not inhibit TGEV infection, while VMP1 KO cells were partially resistant to TGEV infection (S4A and S4B Fig). These results indicated that the VTT-domain protein TMEM41B was more important than the other two genes in TGEV infection. To further determine whether knocking out TMEM41B can inhibit TGEV infection in other porcine cells, we also performed experiments with swine testicle (ST) cells, which were previously demonstrated to be susceptible to TGEV infection; results revealed that TMEM41B KO ST cells were not readily infected by TGEV (S4C Fig). Thus, our CRISPR-based screening strategy discovered numerous candidate host factors related to the replication of the α-CoV TGEV, and we demonstrated that TMEM41B is an essential host factor for TGEV replication.

## TMEM41B is an intrinsic negative regulator of innate immune responses, but this protein does not participate in TGEV replication through the interferon signaling pathway

TMEM41B is predicted to contain six transmembrane domains and to localize to the endoplasmic reticulum (ER). Considering the known function of TMEM41B in the canonical autophagy pathway [34–36], we attempted to determine whether the canonical autophagy pathway is essential for TGEV infection. We generated an autophagy related gene 5 (ATG5) KO cell line, which is an essential molecule for autophagy induction [37]. Our Western blot results showed that ATG5 KO cells produced significantly less lipidated microtubule-associated protein 1A/1B-light chain 3 (LC3)-II upon starvation and aloxistatin (E64d) conditions compared with WT cells, confirming the inability of ATG5 KO cells to initiate autophagy (S5 Fig, middle). However, the ATG5 KO and WT cells remained susceptible to TGEV infection (S5 Fig, right). These findings suggested that the canonical autophagy pathway is not required for TGEV replication.

We subsequently explored the potential downstream signaling pathways affected by the KO of TMEM41B and their potential relationships to the observed inhibition of TGEV replication. RNA sequencing (RNA-seq) was performed with the total RNA isolated from uninfected and

TGEV-infected TMEM41B KO and WT cells (Sheets A-D in S3 Data). Unexpectedly, a comparison of the transcriptomes of TGEV-uninfected WT and TMEM41B KO cells, with more than 1,115 differentially expressed genes (DEGs), indicated that TMEM41B KO cells exhibited altered expression of multiple genes related to several aspects of biological functions (Sheet C in S3 Data). Protein–protein interaction (PPI) network analysis revealed that upregulated genes in TMEM41B KO cells were enriched in "interferon signaling" (S6–S9 Figs). Meanwhile, highly expressed interferon-stimulated genes (ISGs) in TMEM41B KO cells were most significantly associated with interferon signaling pathways according to Gene Set Enrichment Analysis (GSEA) (Fig 3A–3D). The differentially expressed ISGs, such as *MX1*, *OASL*, *IFIT1*, *IFI6*, and *RSAD2*, were then selected to undergo verification via quantitative reverse transcription PCR (RT-qPCR). The relative expression levels of the genes from RT-qPCR were consistent with those from the RNA-seq data (Fig 3E and Sheets A-D in S3 Data).

To determine whether the ISG-activation signature was specific to the loss of TMEM41B function, we performed rescue experiments and found that the mRNA expression of *MX1*, *OASL*, *IFIT1*, *IFI6*, and *RSAD2* in TMEM41B-KO-rescue cells decreased to WT level (Fig 3F). In addition, TMEM41B KO human HEK293T cells and mouse L929 cells were generated by CRISPR/Cas9 technology (S10A and S10C Fig). Compared with TMEM41B KO PK-15 cells, knocking out TMEM41B only slightly increased ISG expression levels in HEK293T and L929 cells, indicating that the ability to upregulate ISGs varies by cell type (S10B and S10D Fig). We further detected the mRNA expression of ISGs in WT cells by stably expressing *TMEM41B* after stimulation with polyinosinic acid–polycytidylic acid (PolyI:C), a synthetic double-stranded RNA (dsRNA) analog which is an immunostimulant that acts as the most potent interferon inducer [38]. Compared with WT cells, TMEM41B overexpression can significantly reduce ISG expression levels, thereby inhibiting the host response to PolyI:C stimulation (S11A–S11E Fig). Subsequently, we investigated whether the interferon signaling pathway is involved in the TGEV replication in TMEM41B KO cells. TMEM41B and IFNAR (interferon-α/β receptor) double PK-15 KO cells were generated. Compared with TMEM41B KO cells, the expression levels of several ISGs (*i.e.*, *RASD2*, *OASL*, and *IFIT1*) in TMEM41B and IFNAR double KO cells were reduced, almost to the same level as those of WT cells (S11F–S11H Fig). However, double KO cells retained a strong ability to inhibit TGEV replication (Fig 3G). Overall, these results revealed that TMEM41B plays an intrinsic negative regulation role in immunity response, which does not inhibit TGEV replication by IFNAR-mediated activation of the interferon signaling pathway.

## TMEM41B KO inhibits the internalization and early-stage replication of TGEV

We next investigated which stages of TGEV replication were affected in TMEM41B KO cells. The absorption rate of virus particles in TMEM41B KO and WT cells at 4°C for 1 h was measured. The result showed that these two cell lines had the same absorption capacity of virus particles (Fig 4A). We found that the positive fluorescence signal of TGEV N protein is evenly distributed on the cell membrane of TMEM41B KO and WT cells, indicating that virus particles can still bind to TMEM41B KO cells (Fig 4B). These results suggested that knocking out TMEM41B does not affect the absorption of virus particles by PK-15 cells. Subsequently, we performed confocal microscopy to investigate viral endocytosis stage. TMEM41B KO or WT cells were first challenged with TGEV (MOI = 50) at 37°C for 30 min, and the expression levels of TGEV N protein were monitored as a reporter for the internalization of virions. The fluorescence signal of the N protein in the cytoplasm of TMEM41B KO cells was significantly reduced compared with those of WT cells (Fig 4C, left); the normalized mean fluorescence

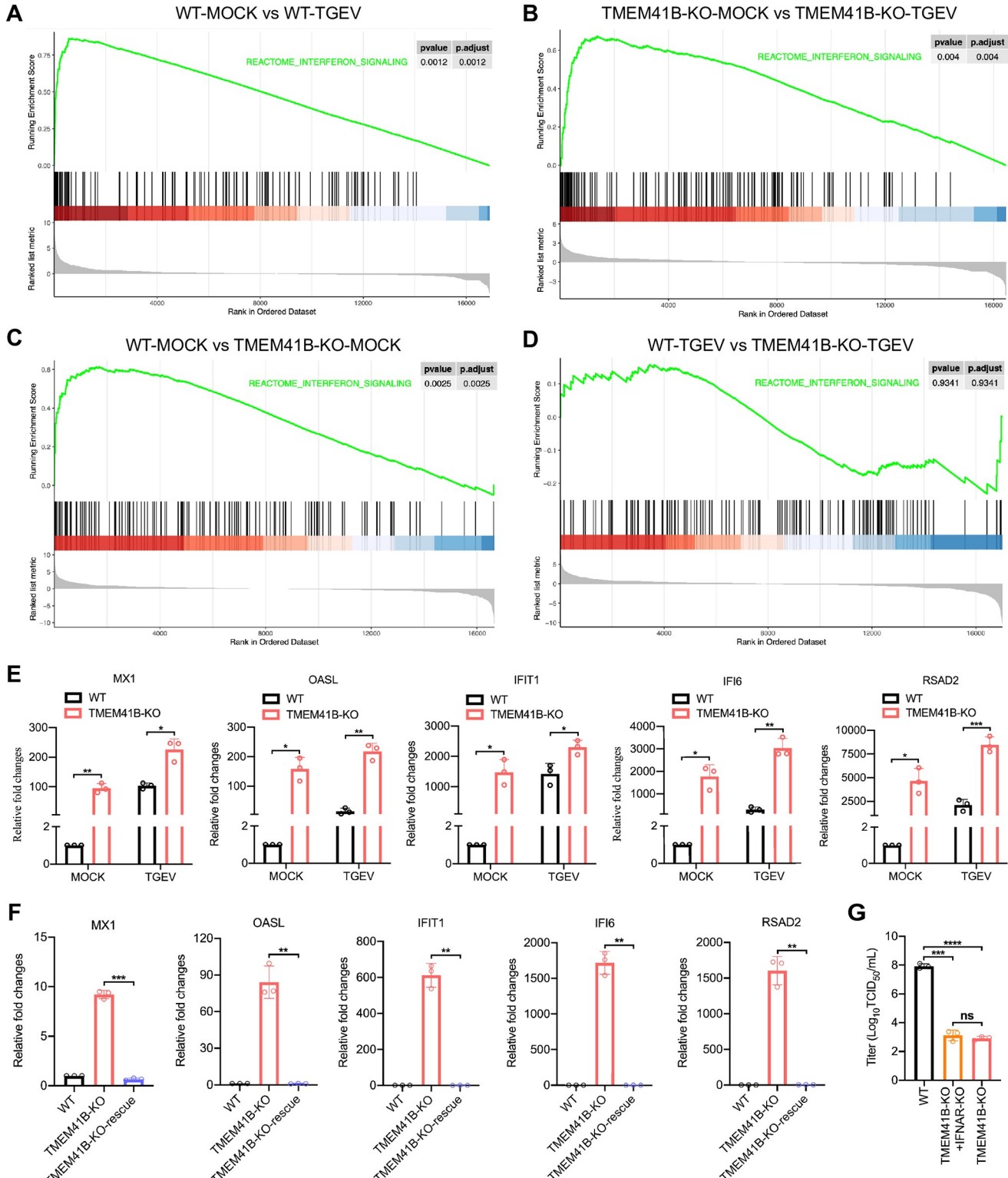

**Fig 3. TMEM41B is an intrinsic negative regulator of innate immune responses. (A-D)** Gene Set Enrichment Analysis (GSEA) for differentially expressed genes from the RNA-seq data shows a distinct upregulation of the interferon signaling pathway across the four pairwise groups **(A)** WT-MOCK vs WT-TGEV; **(B)** TMEM41B-KO-MOCK vs TMEM41B-KO-TGEV; **(C)** WT-MOCK vs TMEM41B-KO-MOCK; **(D)** WT-TGEV vs TMEM41B-KO-TGEV. **(E)** RT-qPCR validation of the mRNA expression of ISGs selected by RNA-seq results. **(F)** RT-qPCR outcome reveal that the stable rescue of TMEM41B in TMEM41B KO cells led to the significant suppression of genes associated with the interferon signaling pathway. **(G)** Viral

titers for WT, TMEM41B-KO and TMEM41B/IFNAR double KO cells infected with TGEV. The TMEM41B/IFNAR double KO cells still maintain the ability to resist TGEV replication compared with TMEM41B KO cells as determined by the TCID$_{50}$ assay. WT, wild-type; KO, knockout; hpi, hours post-infection; MOCK, uninfected cells; TGEV, Transmissible gastroenteritis virus infected cells. *$P$ < 0.05; **$P$ < 0.01; ***$P$ < 0.001. $P$ values were determined by two-sided Student's t-test. Data are representative of at least three independent experiments.

intensity data indicated that the extent of virion internalization decreased by approximately 30% in TMEM41B KO cells (Fig 4C, right). Our transcriptome data (Sheets A-D in S3 Data) and subsequent RT-qPCR validation data for TMEM41B KO and WT cells highlighted that some of the significantly downregulated genes (*e.g.*, *DAB2*, *THRB*) have functions related to endocytosis (Fig 4D). Thus, TMEM41B possibly contributes to TGEV replication by affecting several endocytosis-related processes through which the virus is internalized into cells.

Furthermore, we performed transmission electron microscopy (TEM) to evaluate the effects of knocking out TMEM41B on virus particle assembly. Unlike WT cells, no vesicle-wrapped virus-like particles were produced in TMEM41B KO cells (Fig 4E). The extracellular and intracellular viral titers of TMEM41B KO cells were approximately equal and both lower than those of WT cells (Fig 4F). These results thus suggested that knocking out TMEM41B severely impaired virion production. We found that TMEM41B KO only partially suppressed TGEV internalization (Fig 4C), but this result does not fully explain the observed inhibition of TGEV replication. We therefore examined the early-stage replication of TGEV in TMEM41B KO and WT cells. Confocal microscopy analysis indicated that dsRNA formation (Fig 4G) and TGEV N protein accumulation (Fig 4H) were inhibited almost entirely in TMEM41B KO cells by 3 h post-infection (hpi) upon TGEV infection. Considering that dsRNA acts as the intermediate of CoV replication, these results revealed that TMEM41B is important not only for the internalization stage but also for the early-stage replication of TGEV infection.

## TMEM41B contributes to the formation of CoV ROs

Similar to several other positive-strand RNA viruses of eukaryotes, CoV infection hijacks the intracellular membranes of host cells to form double-membrane vesicles (DMVs) [39–42]. Previous studies revealed that DMVs are a prominent type of viral ROs [43–46]. TMEM41B is an ER transmembrane protein previously reported to be required for phagophore maturation [35]. Considering that parts of the autophagy process show similarities to DMV formation [47–49], we hypothesized that TMEM41B may participate in CoV DMV formation. Thus, we performed TEM assay to evaluate the ability of CoV RO formation in TMEM41B KO and WT cells. We detected typical DMVs (diameter ~200 nm) in WT cells upon TGEV infection (Fig 5A and 5B). Other types of ROs were also found in the TGEV-infected WT cells, including the small open double-membrane spherules (DMSs) and irregular coiled membrane formed by ER expansion (Fig 5C and 5D). The number of DMVs between TMEM41B KO and WT cells were calculated. In the 68 random observation fields of TGEV-infected WT cells by TEM assay, ~35 DMVs with a diameter of ~200 nm were found. By contrast, no obvious DMVs were observed in TMEM41B KO cells (Fig 5E and 5F). We found that numerous vesicles were present in TMEM41B KO cells (Fig 5E and 5F). In contrast with virus-induced organelles, the vesicles were monolayered and had different morphologies with varying sizes. These unidentified vesicles were similar to the previously reported features, which may be the precursor of the autophagosome that originated from the ER membrane [34].

We subsequently co-transfected CoV nonstructural protein 3 (NSP3) and *TMEM41B* into TMEM41B KO cells. Confocal fluorescence results showed that TMEM41B and NSP3 are co-localized (Fig 5G). Previous studies have shown that the CoV NSP3 participates in CoV DMV formation [25,50–52]. Our results thus suggested that TMEM41B may act together with the

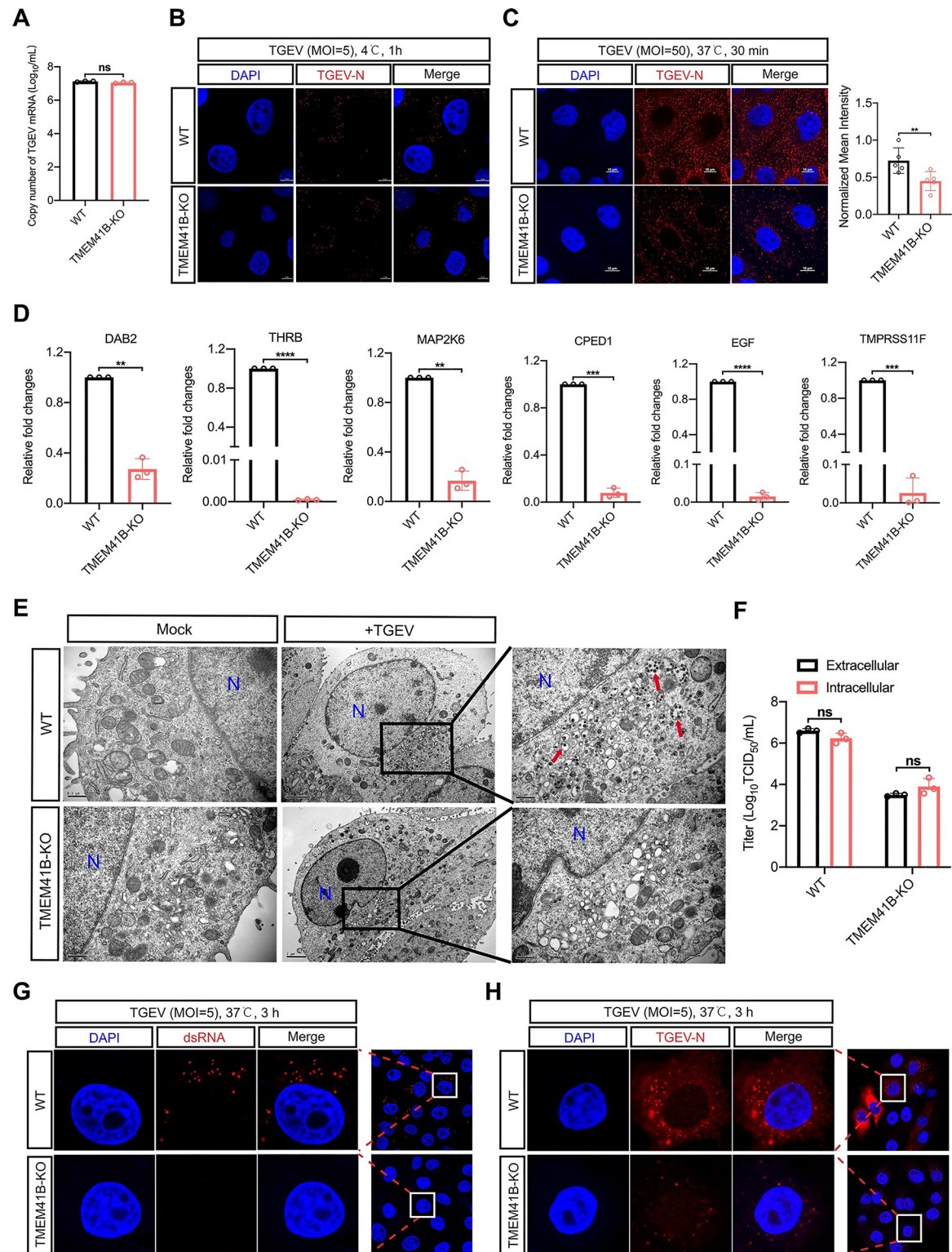

**Fig 4. The internalization and early-stage replication of TGEV were impaired on TMEM41B KO cells. (A)** Evaluation of the adsorption activity of TGEV on TMEM41B KO and WT cells by absolute quantitative real-time PCR assay. WT and TMEM41B KO cells were infected with TGEV (MOI = 5) at 4°C for 1 h and assessed for TGEV adsorption. **(B)** Confocal microscopy assay for detection of TGEV N protein (Red) on WT and TMEM41B KO cells infected with TGEV (MOI = 5) at 4°C for 1 h. Scale bars = 5 μm. **(C)** Evaluation of the TGEV endocytosis stage in TMEM41B KO cells. WT and KO cells were infected with TGEV (MOI = 50) at 4°C for 1 h and

transferred to 37˚C for 30 min. Left: Confocal microscopy assay for detection of TGEV N protein (Red) expression in TMEM41B KO and WT cells. Right: Normalized mean fluorescence intensity was standardized by the ratio of each fluorescence intensity was divided by the maximum fluorescence intensity, n≧3. Scale bars = 10 μm. **(D)** RT-qPCR validation of mRNA expression of endocytic pathway genes enriched by RNA-seq. **(E)** Evaluation of the effects of TMEM41B KO cells on virus particle assembly by transmission electron microscope. Compared with WT cells, no virus-like particles (red arrows) wrapped in vesicles of varying sizes were found in TMEM41B KO cells. Scale bar, 2 μm or 500 nm as indicated. **(F)** Assessment of TGEV release stage of replication in TMEM41B KO and WT cells infected with TGEV (MOI = 5). Intracellular and extracellular viral titers were evaluated by virus $TCID_{50}$ assays at 24 hpi. **(G and H)** Confocal microscopy to evaluate early-stage TGEV replication by detecting **(G)** dsRNA formation and **(H)** TGEV N protein expression in WT and TMEM41B KO cells infected with TGEV (MOI = 5) at 3 hpi. WT, wild-type; KO, knock out; hpi, hours post-infection; MOI, multiplicity of infection; Mock, uninfected cells; TGEV, Transmissible gastroenteritis virus infected cells; N, Nucleus, dsRNA, double-stranded RNA; DAPI, 4',6-diamidino-2-phenylindole. **$P < 0.01$; ***$P < 0.001$; ****$P < 0.0001$; ns, no significant. *P* values were determined by two-sided Student's t-test. Data are representative of at least three independent experiments.

CoV NSP3 in DMV formation. Moreover, subcellular localization of TMEM41B heterologous expression was detected in uninfected or TGEV-infected TMEM41B KO cells. We found that TMEM41B that were expressed in the uninfected KO cells exhibited diffusion in the reticular structure and filling up of the cytoplasm (Fig 5H). By contrast, the fluorescent signal of TMEM41B was significantly accumulated in TGEV-infected KO cells, and the positive signal was co-localized with dsRNA (Fig 5H). We also found that murine TMEM41B co-localized with dsRNA when mouse fibroblasts (L929 cells) were infected with the MHV-A59 strain (Fig 5I). DsRNA was used as a marker for positive-stranded RNA virus replication that always exists in DMVs [46,53]; we thus speculated that the recruitment of TMEM41B to dsRNA sub-cellular sites may be related to CoV replication.

TMEM41B's role in the formation of SARS-CoV-2 ROs was further explored. We found that after co-overexpression of NSP3 and NSP4 from SARS-CoV-2 in WT cells, the ER morphology was visibly swollen and distorted compared with that in mock cells (S12A–S12F Fig). Numerous ER membranes have obvious membrane curvatures, resulting in large deformations (S12E and S12F Fig). The result showed that SARS-CoV-2 non-structural proteins (NSPs) can induce small DMVs with a diameter of ~100 nm in HEK293T cells (S12G–S12L Fig), which coincides with previous reports on Middle East respiratory syndrome coronavirus (MERS-CoV) and SARS-CoV [50]. By contrast, we found that the ER morphology has no obvious swelling or deformation after co-expression of NSP3 and NSP4 in TMEM41B KO cells (S12G–S12L Fig). Lastly, we provided a hypothetical model for TMEM41B's role in CoV replication (Fig 5J) and speculated that this gene may be involved in membrane bending during RO formation.

## TMEM41B is an essential host factor for the replication of numerous viruses from different families

Multiple sequence alignment of human, pig, and mouse TMEM41B orthologs showed striking conservation, especially in the VTT domain (Fig 6A). We hence examined the potential effects of TMEM41B on mice MHV infection in TMEM41B KO and its reconstituted L929 cells. Compared with the WT group, the supernatant virus titers of the TMEM41B KO L929 cells significantly decreased as measured in 50% tissue culture toxicity dose assays ($TCID_{50}$ assay), while the virus replication can be restored on TMEM41B-KO-rescue cells (Fig 6B). We also found that porcine deltacoronavirus (PDCoV) replication was significantly inhibited in TMEM41B KO cells compared with that in WT cells (Fig 6C) as demonstrated by the reduced supernatant virus titers of the TMEM41B KO cells in the $TCID_{50}$ assay.

Additional assays showed that TMEM41B reduced the replication ability of positive-strand RNA viruses, including porcine JEV (Flaviviridae) (Fig 6D). We observed that TMEM41B KO led to significantly inhibited replication of the negative-strand RNA virus, PR8 influenza A

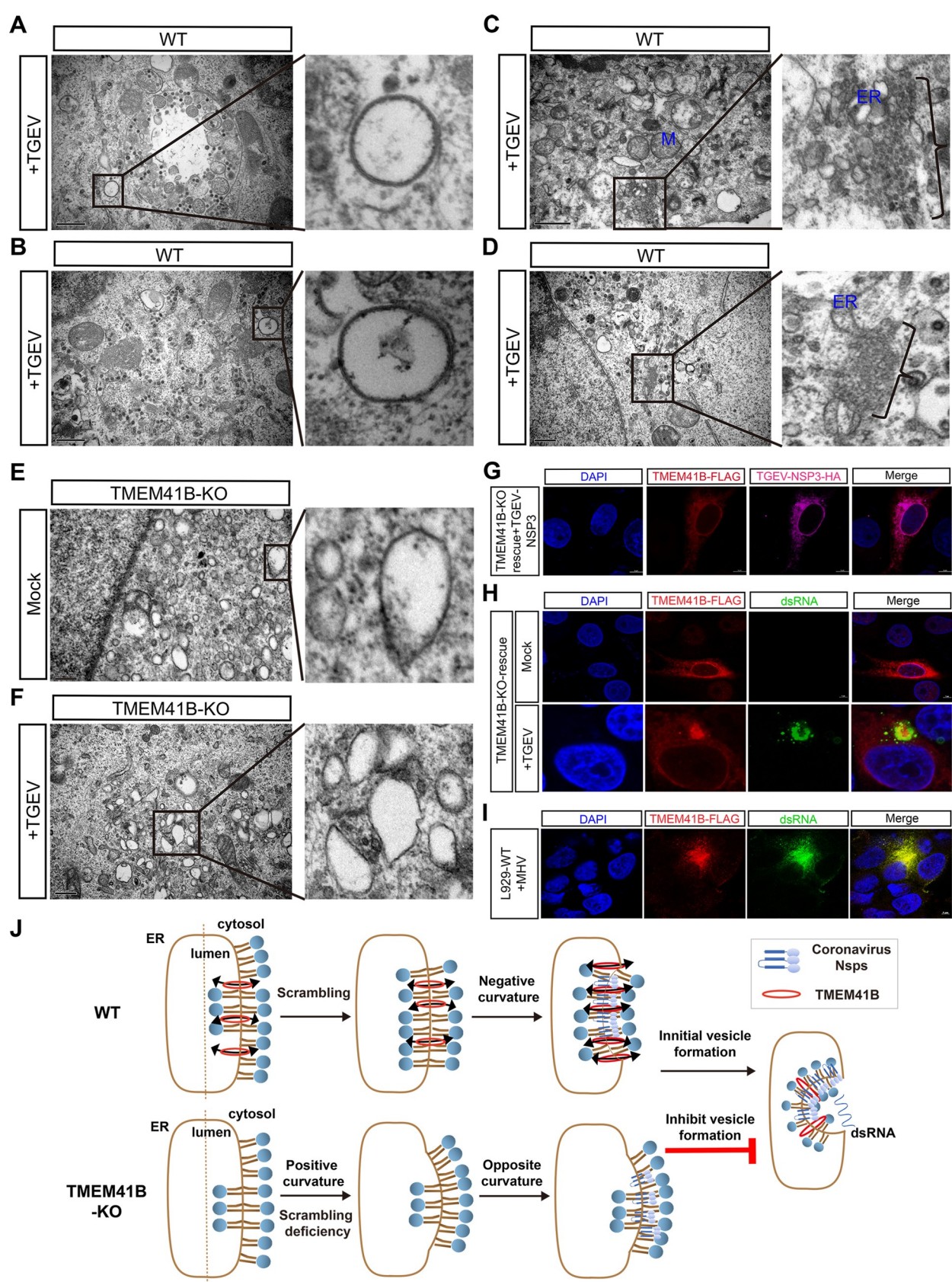

**Fig 5. TMEM41B contributes to the formation of CoVs replication organelles. (A-D)** Representative images from transmission electron microscope of TGEV (MOI = 0.1, 24 h) induces membrane structures. There are typical DMVs (diameter ~200 nm) were observed (**A** and **B**). Electron micrograph of an area with abundant double-membrane spherule (DMSs) (**C**) and irregular coiled membrane formed by ERs expansion (**D**). Scale bar, 1μm or 500 nm. (**E-F**) Electron micrograph showing mock-infected and TGEV-infected (MOI = 0.1, 24 h) TMEM41B KO cells bearing several monolayer vesicles. Scale bar, 500 nm or 250 nm. (**G**) Confocal fluorescence microscopy analysis of the co-localization of TGEV-NSP3-HA (indicated in red) and restored expression of TMEM41B-FLAG (indicated in fuchsia) in TMEM41B KO cells. Scale bar, 5 μm. (**H**) Analysis of the subcellular location of ectopic expression TMEM41B in KO cells mock-infected (above) or infected with TGEV (MOI = 1, 12 h) (bottom). The restored expression of TMEM41B (indicated in red) were co-localization with dsRNA (indicated in green). Scale bar, 5 μm. (**I**) Assessment of the co-localization of dsRNA (indicated in green) and ectopic expression of TMEM41B-FLAG (indicated in red) in L929 cells during MHV (MOI = 1, 12 h) infection via confocal fluorescence microscopy. Scale bar, 5 μm. (**J**) A model for the role of TMEM41B in the formation of coronavirus replication organelles. WT, wild-type; KO, knock out; Mock, uninfected cells; TGEV, Transmissible gastroenteritis virus infected cells; MHV, Mouse hepatitis virus infected cells; M, Mitochondria; ER, Endoplasmic reticulum; DAPI, 4',6-diamidino-2-phenylindole; HA, HA-Tag; FALG, 3×FLAG-Tag; dsRNA, double-stranded RNA.

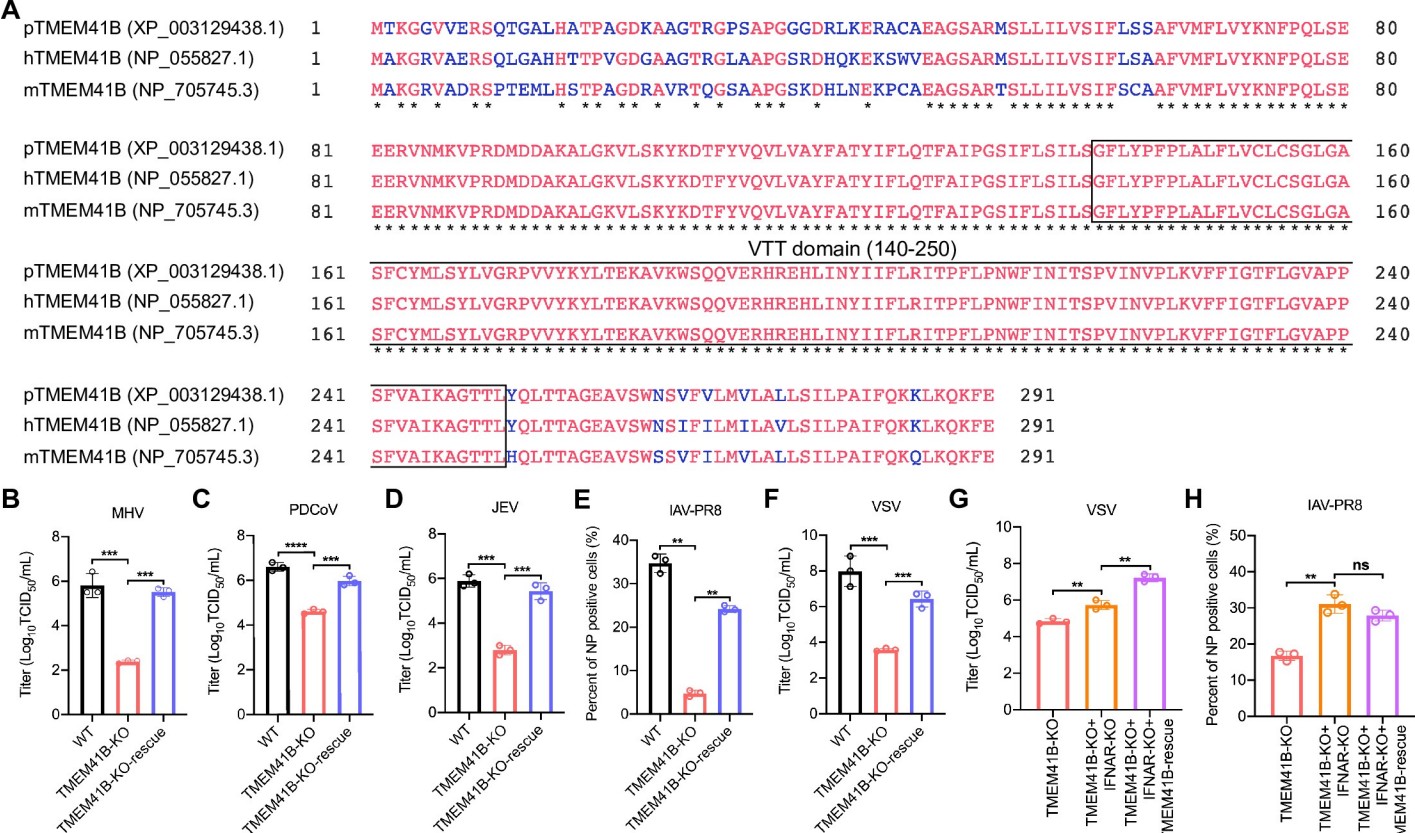

**Fig 6. TMEM41B is a host factor for the replication of multiple viruses. (A)** Alignment of TMEM41B amino acid sequences from pigs, human and mice. The accession Numbers of TMEM41B from NCBI protein database for *Sus scrofa* is XP_003129438.1 (pTMEM41B), for *Homo sapiens* is NP_055827.1 (hTMEM41B), and for *Mus musculus* is NP_705745.3 (mTMEM41B). Protein sequences were aligned using Constraint-based Multiple Alignment Tool (COBALT). The dark star indicates the regions of sequence conservation amongst the three species (letters indicated in red); the blue letters indicate the regions with sequence variation among the three species; the box indicates the VTT domain. (**B**) Rescue assays for WT, TMEM41B-KO and TMEM41B-KO-rescue L929 cells infected with MHV (MOI = 0.1, 24 hpi). Cell supernatant was collected and assessed for MHV titers using $TCID_{50}$. (**C-F**) Rescue assays for WT, TMEM41B-KO and TMEM41B-KO-rescue PK-15 cells infected with (**C**) PDCoV (MOI = 0.01, 18 hpi), (**D**) JEV (MOI = 0.1, 24hpi), (**E**) IAV-PR8 (MOI = 0.01, 24 hpi) and (**F**) VSV (MOI = 0.01, 18 hpi). The cell supernatant was collected and tittered using $TCID_{50}$ assays and immunofluorescence assays to detect NP positive cells. (**G**) Viral titers for TMEM41B-KO, TMEM41B/IFNAR double KO and TMEM41B/IFNAR double KO-TMEM41B-rescue cells infected with VSV (MOI = 0.1, 18 hpi) and cell supernatants was collected and titrated using $TCID_{50}$. (**H**) Immunofluorescence assays for TMEM41B-KO, TMEM41B/IFNAR double KO and TMEM41B/IFNAR double KO-TMEM41B-rescue cells infected with IAV (MOI = 0.1, 18 hpi) and detection the NP positive cells. WT, wild-type; KO, knockout; hpi, hours post-infection; MHV, Mouse hepatitis virus; PDCoV, porcine deltacoronavirus; JEV, Japanese encephalitis virus; IAV-PR8, PR8 influenza A virus; VSV, Vesicular stomatitis virus. $^{**}P < 0.01$; $^{***}P < 0.001$; $^{****}P < 0.0001$. P values were determined by two-sided Student's t-test. Data are representative of at least three independent experiments.

virus (IAV-PR8, H1N1) (Fig 6E) and vesicular stomatitis virus (VSV) (Fig 6F). In addition, TMEM41B KO and TMEM41B/IFNAR double KO and WT cells were infected with VSV. The results showed that the VSV infectious titer in double KO cells was ~1 log higher than that of TMEM41B KO cells. However, rescue TMEM41B in the TMEM41B/IFNAR double KO cells further increased VSV replication compared with that in the TMEM41B/IFNAR double KO cells. This result suggested that the upregulation of ISG level in TMEM41B KO cells only partially inhibit VSV replication, suggesting that TMEM41B may participate in VSV replication through other mechanisms (Fig 6G). In comparison, the replication of IAV increased significantly in the TMEM41B/IFNAR double KO cells compared with that in the TMEM41B KO cells (Fig 6H). Meanwhile, rescue TMEM41B in the TMEM41B/IFNAR double KO cells does not facilitate further IAV replication. This result indicated that the upregulation of ISG level in TMEM41B KO cells may be the main factor inhibiting IAV replication. These results revealed that TMEM41B is an essential host factor for the replication of various RNA viruses.

## TMEM41B is required for CoV infection *in vivo*

To determine whether knocking out TMEM41B affects the ability of α-CoV to replicate *in vivo*, TMEM41B KO mice were generated using CRISPR/Cas9 technology. Two sgRNAs were employed to delete the region spanning Exons 2 to 5 of the *TMEM41B* gene (S13A Fig). However, consistent with other studies reporting TMEM41B as an essential component for mouse embryonic development [54], we were only able to obtain heterozygous TMEM41B KO mice (S13B–S13E Fig). In heterozygous KO mice (TMEM41B$^{+/-}$), the *TMEM41B* mRNA levels were reduced by approximately 60% in liver tissue (S13F Fig). The impairment of autophagic flux characterized by p62 accumulation was significantly increased in the liver and kidney of TMEM41B$^{+/-}$ mice compared with that in WT mice. This result suggested that autophagic activity was disturbed in TMEM41B$^{+/-}$ mice (S13G Fig). In addition, the serum lipid concentrations, including those of triglyceride, total cholesterol, and low-density lipoprotein cholesterol, in the serum of TMEM41B$^{+/-}$ mice decreased significantly compared with those in WT mice (S13H Fig). The results indicated that TMEM41B$^{+/-}$ mice may have defective on lipid transport, which is consistent with a previous study [55].

Subsequently, TMEM41B$^{+/-}$ and WT mice (approximately 4 weeks old, n = 6) were intraperitoneally injected with MHV-A59 (plaque-forming units, p.f.u. = $5 \times 10^5$). The control groups received an equal volume of Dulbecco's Modified Eagles Medium Media (DMEM) by the same route. Each mouse's weight was monitored daily, and lethality was recorded. The body weight change of TMEM41B$^{+/-}$ mice infected with CoV MHV-A59 significantly declined compared with those of WT mice (Fig 7A and 7B). We further found that TMEM41B$^{+/-}$ mice had significantly delayed disease progression compared with WT mice upon MHV-A59 challenge (Fig 7C). Compared with the livers of MHV-infected WT mice, TMEM41B$^{+/-}$ mice significantly alleviated liver tissue necrosis and damage caused by MHV infection (Fig 7D). Histopathological analysis showed extensive necrotic areas in WT mice livers but only focal necrosis loci in TMEM41B$^{+/-}$ mice livers (Fig 7E). Measurement of the viral titers in TMEM41B$^{+/-}$ and WT livers showed that the virus titers of TMEM41B$^{+/-}$ mice livers were significantly reduced compared with those of WT mice livers (Fig 7F). These results revealed that TMEM41B is involved in CoV MHV infection *in vivo*.

## Discussion

Emerging CoVs pose a severe threat to human and animal health worldwide. Additional studies investigating the replication mechanisms of CoVs, which can potentially identify novel targets for developing drugs to treat CoV infections, are urgently needed. The present study used

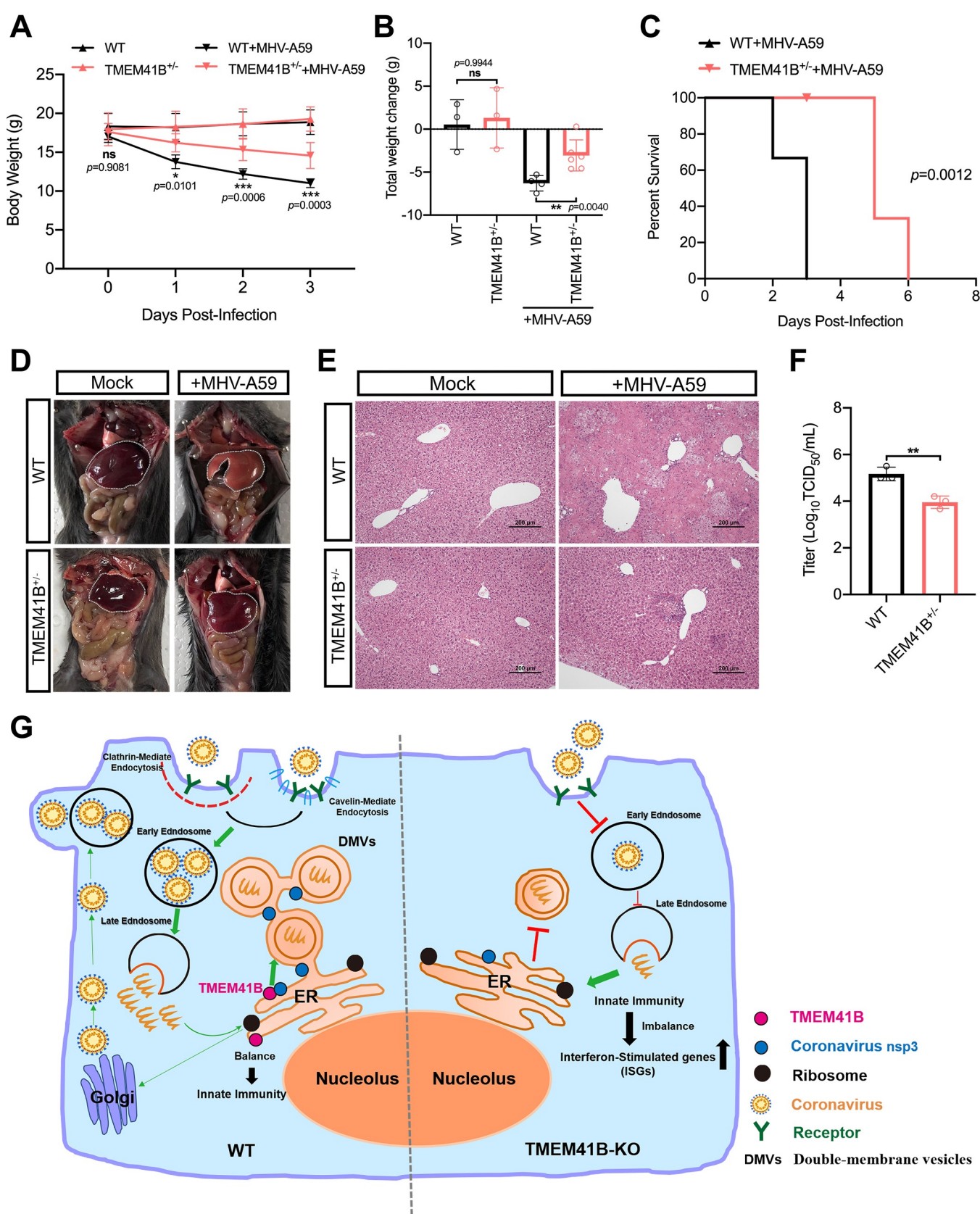

**Fig 7. TMEM41B is required for CoVs infection *in vivo*. (A)** Line graphs showing the body weight of TMEM41B$^{+/-}$ and WT mice during the 3 days post-inoculation with MHV A59 (n = 6 mice per genotype). $^{*}P < 0.05$; $^{**}P < 0.01$; $^{***}P < 0.001$; ns, no significant. *P*-values were determined by two-way measure. **(B)** The total weight change of TMEM41B$^{+/-}$ and WT mice during the 3 days post-inoculation of MHV A59 (n = 6 mice per genotype). $^{**}P < 0.01$; ns, no significant. *P*-values were determined by one-way measure. **(C)** Survival curves for TMEM41B$^{+/-}$ and WT mice infected with MHV A59 (n = 6 mice per genotype). **(D and E)** Histopathological analysis of the degree of liver damage in TMEM41B$^{+/-}$ and WT mice infection with MHV A59 at 3 dpi. **(D)** Gross postmortem examination of liver tissue; **(E)** histological examination of Hematoxylin-eosin (H&E) stained liver tissue. **(F)** The virus titers of half livers of TMEM41B$^{+/-}$ and WT mice were measured TCID$_{50}$ assay with L929 cells (n = 3 mice per genotype). $^{**}P < 0.01$. *P* values were determined by two-sided Student's t-test. **(G)** A model illustrating the roles of TMEM41B in the CoV replication cycle. In WT cells, CoV binds to its receptor and enters through clathrin- and caveolin-mediated endocytosis before releasing their genome into the cytoplasm after membrane fusion in the early/late endosome. Subsequently, CoVs Nsps (NSP3, NSP4, and NSP6) act with TMEM41B and other transmembrane host proteins in the formation of ROs. In TMEM41B KO cells, endocytosis is impaired, and the internalization of virions was reduced. Moreover, the formation of DMVs was blocked during membrane elongation leading to the inhibition of TGEV replication so that almost no virions were produced. DMVs, double-membrane vesicles; ER, Endoplasmic reticulum; Mock, uninfected mice; MHV-A549, A549 Mouse hepatitis virus infected mice. WT, wild-type; KO, knockout; TMEM41B$^{+/-}$ mice used in Fig 7 were the F2 generation. Data are representative of at least three independent experiments. Source data are provided as a Source Data file. The experiments were repeated three times with similar results and representative results shown. Scale bar, 200 nm.

a PigGeCKO library and identified host factors involved in TGEV replication in host cells. The known cell-surface receptor of TGEV (*i.e.*, *ANPEP*) was among the identified factors [31,32]. We also screened candidate host factors, including *TMEM41B* and *DYRK1A*, both of which were also reported from SARS-CoV-2 screenings [27,28,56]. Moreover, experimental validation of hits from our CRISPR screen demonstrated that numerous candidate host factors participated in TGEV replication. We found that TMEM41B functions in the replication of α-, β-, and δ-group CoVs. Several studies have shown that TMEM41B is involved in the replication of members of the human CoVs, such as SARS-CoV-2, HCov-229E, and HCov-NL63 [27,28,56,57]. However, the regulatory mechanism of TMEM41B in CoV replication remains unclear. In the present study, we systematically studied the possible conservative mechanism of TMEM41B in CoV replication. Moreover, our animal experiments demonstrated the functional relevance of TMEM41B for CoV infection *in vivo* for the first time. We found that upon MHV-A59 infection, TMEM41B$^{+/-}$ mice had significantly delayed disease progression and had significantly reduced virus titers.

We found that TMEM41B confers cell susceptibility to virus entry (Fig 4A–4C) and cell permissively for CoV replication (Fig 4G and 4H). Although virion internalization level on TMEM41B KO cells were reduced, numerous virions successfully internalized into the cytoplasm (Fig 4C). However, we found that the internalized virions were unable to further replicate in TMEM41B KO cells (Fig 4G and 4H). The results strongly indicated that the replication stage of the virus depends the most on TMEM41B and may be the most biologically relevant stage. Our result was consistent with that of another report that showed that SARS-CoV-2 replication in cells lacking TMEM41B was reduced by more than 80% when the entry step was bypassed [28]. Moreover, the TEM assay result showed that knocking out TMEM41B to inhibit CoV replication was related to CoV DMV formation (Fig 4A–4F).

DMVs offer a conducive microenvironment for viral RNA synthesis and shield viral RNA from innate immune sensors that are activated by dsRNA [22]. Although the viral factors involved in the formation of CoV DMVs are already known, the host factors involved in DMV formation remain largely unknown. Here, we found that TMEM41B can co-localize with CoV NSP3 (Fig 5G), and in response to CoV infection, the subcellular location of TMEM41B co-localizes with dsRNA, thereby participating in CoV replication. These results suggested that TMEM41B is recruited to participate in virus replication, potentially via interactions with NSPs of CoVs. A previous study demonstrated that TMEM41B co-localizes and co-immuno-precipitates with the nonstructural protein 4A (NS4A) and NS4B proteins of yellow fever virus and Zika virus; TMEM41B was found to be recruited to flavivirus RNA replication complexes to facilitate membrane curvature [58]. The present study supported the hypothesis that with the lack of TMEM41B, CoVs NSPs do not have sufficient membrane curvature to generate the

DMVs (S12 Fig). Thus, we suggest that TMEM41B may represent a conserved host factor for the replication of CoVs and other positive-strand RNAs.

In the process of revising the present paper, we found several studies that revealed that TMEM41B acts as a phospholipid scramblase mediating trans-bilayer shuttling of phospholipids [55,59,60]. Combined with our new findings, we speculate that TMEM41B's role in DMV formation is as follows (Fig 5J): TMEM41B may mediate the phospholipids synthesized on the cytoplasmic leaflets and shuttle to the luminal leaflets to promote the phospholipid bilayer's balance. When cells are infected by a CoV, the transmembrane proteins of NSPs (*i.e.*, NSP3, NSP4, and NSP6) encoded by the virus located in the ER are oligomerized and recruit TMEM41B to increase membrane curvature, thereby leading to DMV formation. However, TMEM41B's defect can prevent the CoV-encoded NSPs from modifying the ER membrane curvature to generate DMV (Fig 5J). Overall, the ER scramblase TMEM41B can be recruited to promote ER deformation to consequently promote CoV RO formation.

Interestingly, we found that TMEM41B is an intrinsic negative regulator of cellular innate immunity, which has not been reported in other papers. A previous study reported TMEM41B as an ER-resident protein that localizes at mitochondria-associated ER membranes (MAMs) [54], which are known to be functionally specialized contact sites; MAMs have been recognized as necessary for diverse biological processes, including calcium homeostasis, autophagy [61], and the mediation of intracellular immune synapses which direct antiviral innate immunity [62]. The functions of TMEM41B are thus related to intracellular innate immunity and may be related to its localization at MAMs. In addition, autophagy was found to be a homeostatic process with several effects on immunity [63]. Autophagy represents an anti-inflammatory mechanism in its engagement with inflammasomes, and their substrates protect against endomembrane damage triggered by various agents of endogenous or infectious origin and prevent unnecessary or excessive inflammation [64,65]. The absence of autophagy may also amplify DDX58/RIG-I-like receptor signaling through increased mitochondrial antiviral signaling protein levels on accumulating mitochondria [66]. Therefore, the impaired autophagy pathway on TMEM41B KO cells may trigger excessive innate immunity.

Perhaps the most striking finding from the present study is that TMEM41B KO cells displayed broad-spectrum antiviral effects on other viruses, including the positive-strand RNA virus JEV and the negative-strand RNA viruses IAV and VSV. Our results revealed that IAV is highly sensitive to the upregulated expression of ISGs in TMEM41B KO cells. However, knocking out TMEM41B not only inhibits VSV infection by regulating ISGs. The VSV infection requires normal host lipid metabolism [67]. Several studies have shown that lipid transport is impaired in TMEM41B KO cells [36,57,68]. Whether knocking out TMEM41B may inhibit lipid transport and thereby inhibit VSV infection remain unclear.

In summary, we proposed a hypothetical model to illustrate TMEM41B's roles in the CoV replication cycle. As shown in Fig 7G, after CoV infects host cells and binds to the receptor, CoV enters the cell through endocytosis and then releases its genome into the cytoplasm after early/late endosomal membrane fusion. Subsequently, CoV NSP3 acts with TMEM41B and other transmembrane host proteins in DMV formation. By contrast, the endocytosis of TMEM41B KO cells is impaired, resulting in a reduction in internalized CoV particles. In addition, DMV formation in TMEM41B KO cells is blocked, resulting in the inhibition of CoV replication in host cells. Our findings collectively indicated that TMEM41B is a *bona fide* host factor for CoV replication and emphasized the strong potential of TMEM41B as a therapeutic target for developing (possibly broad-spectrum) drugs to inhibit the replication of emerging and re-emerging CoVs.

## Materials and methods

### Ethics statement

All experiments involving mice were performed in accordance with the recommendations in the Guide for the Care and Use of Laboratory Animals of the Ministry of Science and Technology of China and were approved by the Scientific Ethics Committee of Huazhong Agricultural University (permit number: HZAUMO-2020-0051). Animal care and maintenance protocols complied with the recommendations detailed in the Regulations for the Administration of Affairs Concerning Experimental Animals crafted by the Ministry of Science and Technology of China.

### Plasmid construction

To construct the lentivirus sgRNA expression vector, the lenti-sgRNA-EGFP vector [30] was digested by the *Bbs*I restriction enzyme. The paired oligonucleotides of sgRNA were annealed and cloned into the linearized vector. To construct the overexpression vector for the rescue experiments, the coding sequences of TMEM41B were cloned into the pcDNA3.1(+) vector (Invitrogen) which was linearized with *Nhe*I and *Kpn*I by In-Fusion HD Cloning Plus (Clontech). To construct the overexpression vector for the generation of TMEM41B stable cell lines, the coding sequences of TMEM41B were cloned into the pLVX-T2A-mCherry-Puro vector (Clontech) which was linearized with *Xho*I and *BamH*I. Subsequently, to abolish cleavage by sgRNA and Cas9 in TMEM41B KO cells, a specific point mutation in protospacer adjacent motif sequences, which does not alter the amino acid, was introduced into the TMEM41B coding sequences to construct its mutant overexpression vectors. Sanger sequencing (Tsingke) was performed to confirm all plasmids. All primer sequences are listed in S4 Data.

### Cell culture and transfection

PK-15, HEK293T, and BHK-21 cell lines were purchased from the Cell Bank of the Chinese Academy of Sciences (Shanghai, China). ST (CRL-1746) and L929 (CCL-1) cell lines were purchased from ATCC (USA). All cell lines were then subjected to mycoplasma detection. For cell culture experiments in the present study, all cells were maintained in DMEM supplemented with 10% fetal bovine serum (FBS), 100 U/mL penicillin, 100 μg/mL streptomycin, and incubated at 37°C with 5% $CO_2$. All cell transfections were performed with JetPRIME (PolyPlus) reagent according to the manufacturer's instructions. WT and candidate gene KO PK-15 cells with approximately 80% confluency were first seeded into 12-well plates and transfected with 1 μg plasmid DNA. At 24 h after transfection, cells were incubated with TGEV. At 8 h after incubation, the inoculum was removed and replaced with 1 mL of fresh DMEM containing 10% FBS and 1% penicillin–streptomycin. After 24 h of incubation, immunofluorescence assays and RT-qPCR assays were conducted.

### Viruses

The following viruses were used: TGEV WH-1 strain (GenBank accession numbers: HQ462571.1), MHV A59 strain (GenBank accession numbers: MF618253.1), IAV PR8 strain (GenBank accession numbers: AGQ48042.1), and PDCoV strain CHN-HN-2014 (GenBank accession number: KT336560). VSV and Pseudorabies virus (PRV) were provided by Prof. Gang Cao, and JEV P3 strain was obtained from Prof. Min Cui at Huazhong Agricultural University. TGEV, PSV, VSV, and PRV were propagated in PK-15 cells. JEV and IAV were propagated in Baby Hamster Kidney Fibroblast Cells (BHK-21 line) and chicken embryos, respectively.

## Focused-CRISPR library design and construction

A focused-CRISPR library was designed on the basis of the results of three TGEV challenge rounds of genome-wide CRISPR screening. First, 6–7 sgRNAs were designed against each enriched candidate gene from the top 79 ranked sgRNAs from the third and fifth TGEV challenge rounds (S2 Data). Next, the focused-CRISPR library was synthesized using CustomArray 12K arrays (CustomArray Inc.). Oligonucleotide pools were amplified by PCR using Phusion High-Fidelity PCR Master Mix with HF Buffer (NEB). The PCR products were purified and cloned into the linearized lenti-sgRNA-EGFP vector via Gibson assembly approach. To ensure sufficient coverage, more than $2 \times 10^5$ transformants were collected to generate a plasmid library of sgRNAs.

## sgRNA library lentivirus production

Co-transfection of 12 μg of the PigGeCKO or focused-CRISPR library plasmid, 4 μg of pMD2. G plasmid (Addgene, #12259), and 8 μg of psPAX2 (Addgene, #12260) plasmid per 100-mm dish was performed by using JetPRIME (PolyPlus) according to the manufacturer's instructions. At 60 h after transfection, the cell supernatants were collected, filtered by using a 0.45 μm low-protein binding membrane (Millipore), and then centrifuged at 30,000 rpm and 4°C for 2.5 h. The lentivirus pellets were resuspended in DMEM solution, aliquoted, and stored at −80°C.

## Construction of PigGeCKO cells and focused-CRISPR KO cells and TGEV challenge

The PigGeCKO cells were generated by a total of $\sim 2 \times 10^8$ PK-15-Cas9 cells and infected with the genome-wide sgRNA library lentiviruses at an MOI of 0.3 as described in a previous publication [30]. In addition, the focused-CRISPR KO cells were generated by a total of $\sim 2 \times 10^7$ PK-15-Cas9 cells and infected with the focused-CRISPR sgRNA library lentiviruses. To examine the coverage of PigGeCKO cells and focused-CRISPR KO cells, genomic DNA (gDNA) from a total of $\sim 7 \times 10^6$ cells were extracted using Blood & Cell Culture DNA Midi Kit (QIAGEN). For the genome-wide CRISPR screening assays, $\sim 6 \times 10^7$ PigGeCKO cells were infected with TGEV at an MOI of 0.001 in DMEM without FBS and incubated at 37°C and 5% $CO_2$. For focused-CRISPR screening assays, $\sim 1 \times 10^7$ focused-CRISPR KO cells were infected with TGEV at an MOI of 1. After 2 h of incubation, the inoculum was removed and replaced with fresh DMEM supplemented with 2% FBS and 1% penicillin–streptomycin. At 7 days after infection, surviving cells were collected and expanded for deep sequencing analysis and the next round of infection.

## Illumina sequencing of sgRNAs in the PigGeCKO cells and focused-CRISPR KO cells

The gDNA from the surviving PigGeCKO cells and focused-CRISPR KO cells were first extracted using a Blood & Cell Culture DNA Midi Kit (QIAGEN). The sgRNA-coding region was amplified by PCR using Q5 Hot Start High-Fidelity DNA Polymerase (NEB). PCR products were purified using MinElute PCR Purification Kit (QIAGEN) followed by Illumina HiSeq 3000 Next-generation Sequencing. Mapped read counts were subsequently used as input for the MAGeCK analysis software package (Version 0.5) [69]. The top 0.5% ranked sgRNAs from each TGEV challenge round were then used to identify enriched targeted protein-coding genes. All primers are listed in S4 Data.

## Generation of candidate gene KO cell lines

sgRNA targeted candidate genes were cloned into the linearized lenti-sgRNA-EGFP, and lentivirus containing sgRNAs were produced as described in a previous study [30]. The sgRNA lentivirus was then transduced into PK-15-Cas9 cells. On the third day after transduction, cells with GFP expression were enriched by fluorescence-activated cell sorting. Editing efficiency was examined using Tracking of Indels by Decomposition (TIDE) program [70]. All primers are listed in S4 Data.

## RT-qPCR

Total RNA from cells and viral RNAs from cell suspensions were extracted with the TRIzol Reagent (Invitrogen). Complementary DNAs (cDNAs) were synthesized using PrimeScript RT Reagent Kit with gDNA Eraser (TaKaRa) in a total volume of 10 μL. Each RT-qPCR reaction was carried out with 100 ng of cDNA and 5 nM primer pairs by using SYBR Green Mix (Bio-Rad, USA). The results were monitored using a CFX96 Real-Time PCR Detection System (Bio-Rad, USA) programmed for one cycle of 15 min at 95˚C, followed by 39 cycles of 10 s at 95˚C and 30 s at 60˚C. The relative expression levels were calculated using $2^{-\triangle\triangle Ct}$ method. Beta-actin gene was used as a normalization control. For absolute RT-qPCR, approximately 1 μL of viral RNAs was used as template to synthesize cDNAs. Absolute RT-qPCR assays were performed using SYBR Green Mix and primers specific for the N gene of TGEV in a final reaction volume of 10 μL. The TGEV N protein-coding cDNA sequence from GenBank (accession number: ADY39745.1) was cloned into the pcDNA3.1 vector and used as an internal reference for the quantification of TGEV copy numbers. All primers used in quantitative PCR are listed in S4 Data.

## Virus plaque assay and virus titers

Crystal violet assays were performed as follows: WT and TMEM41B KO PK-15 cells were first incubated in a 24-well plate with different MOIs of TGEV at 37˚C with 5% $CO_2$. The plates were incubated for 1 h and swirled every 15 min to ensure complete infection, after which the inoculum was removed and the cells were overlaid with 50% 2× DMEM, 50% Low Melting Point Agarose (Gibco), 10% FBS, and 1% penicillin–streptomycin for 2 days. The plates were then fixed with 4% formaldehyde for 1 h and stained with 1% crystal violet. Virus titers were performed as follows: Confluent monolayers of different candidate gene KO cells and control cells in 12-well plates were inoculated with TGEV in triplicate at an MOI of 1. The cell supernatants were harvested at 24 hpi, and the virus titers were determined by the $TCID_{50}$ assays using PK-15 cells.

## Immunofluorescence assay

The protein expression levels of TGEV N protein in candidate gene KO cells and control cells were determined via an immunofluorescence assay. Cells grown on the 12-well cell culture plates were first infected with TGEV at different MOIs. At 24 hpi, cells were fixed with 4% paraformaldehyde at room temperature for 30 min and then permeabilized at room temperature for 10 min with cold 0.3% TritonX-100 in phosphate buffered saline (PBS). Cells were incubated with TGEV N antibody (A rabbit anti-TGEV N protein polyclonal antibody was prepared in our laboratory) or anti-dsRNA antibody (SCICONS, #10010200, 1:1,000) at 4˚C overnight, and the primary antibodies were recognized by Alexa Fluor 594 Goat anti-Mouse IgG (H+L) (Invitrogen, #A-11005, 1:1,000), Alexa Fluor 594 Anti-Rabbit IgG (H + L) (Invitrogen, #A-11012, 1:1,000), or Alexa Fluor Plus 647 Goat anti-Mouse IgG (H+L) (Invitrogen,

#A32728, 1:1,000). Cell nuclei were counterstained with 4', 6-diamidino-2-phenylindole (DAPI) (Sigma, #D9542) at room temperature for 1 min in the dark. Cells were observed and imaged with a fluorescence microscope (Thermo Fisher Scientific EVOS FL Auto). The proportion of positive cells were calculated using Image J software (three independent wells were imaged, and one random field of view per well was captured for each experimental phase).

## Immunoblotting assay

To detect the presence of TGEV N protein, total cell lysates were prepared by lysing WT and TMEM41B KO PK-15 cells in 10× sample buffer. Proteins were separated by SDS-PAGE and transferred onto polyvinylidene fluoride membranes (Millipore). Membranes were blocked using 5% skim milk powder (Merck) in PBS with 0.1% Tween-20 (Bio-rad) and blotted with an antibody against TGEV N protein (1:1,000). GAPDH antibodies (Beyotime, #AF5009, 1:3,000) were used as an internal loading control. The primary antibodies were detected with horseradish peroxidase conjugated goat anti-rabbit IgG (Abclonal, #AS014, 1:3,000) or goat anti-mouse IgG (Beyotime, #A0216, 1:1,000), and the secondary antibodies were visualized by ECL Prime Western Blotting Detection Reagents (GE Healthcare, UK).

## MTS and EdU cell proliferation assay

To assess cell proliferation, MTS and EdU cell proliferation assays were performed. The MTS assay was performed using MTS Assay Kit (Abcam, #ab197010). TMEM41B KO PK-15 cells and control cells were seeded into 96-well plates and incubated for 12 or 24 h. Afterward, 20 μL/well MTS reagent was added into each well and incubated at 37˚C for 4 h in standard culture conditions. The plate was briefly shaken, and the absorbance of treated and untreated cells was measured using a plate reader at an optical density (OD) of 490 nm.

For the EdU assay, TMEM41B KO PK-15 cells and control cells were seeded into 24-well plates. After 24 h of incubation, the EdU cell proliferation assay was performed using Beyo-Click EdU Cell Proliferation Kit (Beyotime, #C0075S) with Alexa Fluor 555 according to instruction. The cell nuclei were stained with DAPI (Beyotime, #C1005) at room temperature for 10 min in the dark. Stained cells were visualized under a fluorescence microscope. The proportion of EdU-positive cells was calculated using Image J software (three independent wells were imaged, and one random field of view per well was captured for each experimental phase).

## TEM assay

TMEM41B KO and WT PK-15 cells were infected with TGEV at an MOI of 1 for 12 h. The cells were washed twice with pre-cooled PBS and fixed by adding 3 mL of 2.5% glutaraldehyde (Servicebio) at room temperature for 2 h. After fixation, cells were scraped and transferred into a 2 mL centrifuge tube. Negative-staining electron microscopy was performed by Service-bio company, and images were taken using an HZAU transmission electron microscope platform (HITACHI, #H-7650).

## RNA sequencing and transcriptome analysis

For high-throughput RNA-seq, RNA libraries were created from each group, such as TMEM41B KO cells (*i.e.*, TMEM41B_KO_MOCK, TMEM41B_KO_TGEV) and WT cells (*i.e.*, WT_MOCK, WT_TGEV). Three replicates were conducted for each sample. poly(A) + RNA isolation, library construction, and sequencing were performed by the sequencing platform of the National Key Laboratory of Crop Genetic Improvement using MGISEQ-2000RS.

Sequence quality for all samples was assessed using FastQC (v0.11.7), and quality trimming was conducted using FASTX-Toolkit (v 0.0.14) to remove bases with a Phred33 score of less than 30 while retaining the resulting reads of at least 50 bases in length. The quality trimmed reads were mapped against the reference genome of *Sus scrofa* (v11.1) using HISAT2 (v2.1.0). Gene expression profiling was performed on the basis of the number of reads. Fragments per kilo base of exon model per million mapped reads (FPKM) values of each unigene were used to estimate the expressed values and transcript levels by using SAMtools (v1.7) and HTSeq-count (v0.9.1). Differentially expressed genes (DEGs) were obtained by DESeq2 (v1.30.1) with a *P*-value cutoff $\leqq$ 0.05 and an absolute fold-change of $\geqq$ 1. Subsequently, DEGs in different pairwise groups were analyzed by PPI network (https://string-db.org/) and GSEA [71]. Furthermore, the PPI network of the selected DEGs was visualized using Cytoscape [72].

## PolyI:C stimulation assay

For the PolyI:C stimulation assay, $2 \times 10^5$ PK-15-TMEM41B cells and control cells were incubated overnight in 500 μL cell culture media in 24-well plates. Cells were then incubated with fresh culture media containing PolyI:C (10 μg/mL, InvivoGen, #Tlrl-pic). At 24 hpi, the total RNA from cells was extracted based on previous measure. Relative expression levels of ISGs were detected by RT-qPCR and calculated using the $2^{-\triangle\triangle Ct}$ method. All primers used are listed in S4 Data.

## Confocal microscopy

To observe the endocytosis of the TGEV in host cells, the same amount of TMEM41B KO cells and control cells were cultured in 35-mm Petri dishes overnight. The equivalent dose of TGEV (MOI = 50) was added to each well and incubated at 4˚C for 60 min for complete adsorption and then transferred to 37˚C and incubated for 30 min for endocytosis. Immunofluorescence assays were performed as described above, while the images were acquired using a laser scanning confocal microscope (Nikon). The subcellular localization of TMEM41B in TMEM41B KO cells and control cells transfected with TMEM41B-FLAG-pCDNA3.1 plasmid was observed after 24 h. Afterward, TGEV (MOI = 1) was added into these cells, incubated for 24 h, immunolabeled with a FLAG-tag antibody (Proteintech, #20543-1-AP; MBL, #PM020) and dsRNA antibody, and imaged to identify double fluorescent positive cells. The co-localization of TMEM41B and TGEV NSP3 in TMEM41B KO cells was co-transfected with TMEM41B--FLAG-pCDNA3.1, and TGEV-NSP3-pCAGGS plasmids for 24 h was also observed. These cells were immunolabeled with anti-FLAG-tag and anti-HA-tag antibodies (Proteintech, #66006-1-Ig) and imaged for double-positive fluorescent cells.

## Generation of TMEM41B KO mice and MHV challenge

The TMEM41B KO mice in the C57BL/6 genetic background were generated by Cyagen (China). To generate KO mice, a gDNA fragment spanning Exons 2 to 5 of TMEM41B was deleted by co-injecting sgRNAs and Cas9 mRNA into fertilized mouse eggs. The founders were genotyped by PCR amplification of gDNA extracted from mouse tails, followed by Sanger sequencing. WT and TMEM41B KO mice (approximately 4 weeks old, n = 6) were subjected to intraperitoneal injection with MHV-A59 (p.f.u. = $5 \times 10^5$). The control group received an equal volume of DMEM via the same strategy. The weight of each mouse was monitored daily, and the lethality was recorded. The mock and virus-infected mice were sacrificed at the indicated days after infection to observe the liver's pathology. Liver sections were fixed in buffered formalin (4%) solution, embedded in paraffin, and sectioned into 4 μm tissue sections. Sections were stained with Hematoxylin and Eosin, examined, and imaged under a light

microscope. Livers were homogenized in 1 mL of DMEM and centrifuged at 8,000 rpm for 10 min at 4˚C. The virus titers in the supernatants were collected and measured via $TCID_{50}$ assays with PK-15 cells. P62 protein expression was detected through the following steps: 0.1 g livers and kidney tissues of $TMEM41B^{+/-}$ and WT mice were homogenized in 1 mL RIPA Lysis Buffer (Beyotime, P0013), and the homogenate was crushed by ultrasonication for 1 min. The homogenate was centrifuged at 12,000 rpm for 30 min at 4˚C. Subsequently, the supernatant was collected for Western blot.

## Supporting information

**S1 Fig. Determination of the optimal infection dose of TGEV-induced PK-15 cell death.** The red boxes indicate TGEV-induced cytopathic effects (CPE) in PK-15 cells infected with TGEV at MOIs of 0.001, 0.01, 0.1, or 1. The mock was non-infected TGEV cells used as a negative control. Scale bar, 200 μm. MOI, multiplicity of infection; dpi, days post-infection; TGEV, Transmissible gastroenteritis virus.
(TIF)

**S2 Fig. DNA sequence analysis showing the presence of the mutation in clonal KO cells of _TMEM41B_ (A), _ANPEP_ (B), _LPP_ (C), _TAX1BP1_ (D), and _BARHL2_ (E).** The underline indicates the deleted bases in the KO cells. The black box indicates the PAM sites. PAM: protospacer adjacent motif; sgRNA, small guide RNA; WT, wild-type; KO, knockout; del, deletion; bp, base pairs.
(TIF)

**S3 Fig. Knockout of TMEM41B does not affect the proliferation of PK-15 cells.** (A) Design of CRISPR-resistant pTMEM41B sequences. To abolish cleavage by sgRNA and Cas9 in TMEM41B KO cells, a specific point mutation in PAM sequence, which does not alter the amino acid was introduced into the TMEM41B coding sequences. (B) There was no significant change in the proportion of EdU positive cells in clonal TMEM41B KO cell lines compared to WT cells. Left: representative pictures. Right: quantification of EdU positive cells. Quantitative analysis of immunofluorescence was used by Image J software. (C) TMEM41B KO and PK-15 WT cells were seeded into 96-well plates to evaluate cell proliferation by MTS assays at 12 hpi and 24 hpi. Scale bar, 200 μm. Data are represented as means ± S.D.; n = 3, ns: no significant. pTMEM41B, porcine TMEM41B gene; WT, wild-type; KO, knockout; PAM, protospacer adjacent motif; hpi, hours post-infection; DAPI, 4',6-diamidino-2-phenylindole. ns, no significant. _P_ values were determined by two-sided Student's t-test.
(TIF)

**S4 Fig. TMEM41B KO ST cells show inhibited TGEV replication.** (A) WT cells and VMP1 KO PK-15 cells were infected with TGEV (MOI = 1) for 24h and used for immunofluorescence assays. Left: representative pictures. Right: quantification of TGEV N positive cells. Quantitative analysis of immunofluorescence was conducted with Image J software. Scale bar, 400 μm. (B) WT cells and TMEM41A KO PK-15 cells were infected with TGEV (MOI = 1) for 24h and used for qRT-PCR to measure the TGEV N copy number. (C) WT cells and TMEM41B KO ST cells were infected with TGEV (MOI = 1) for 24 h and used for qRT-PCR to measure TGEV N copy number. WT, wild-type; KO, knockout; DAPI, 4',6-diamidino-2-phenylindole; ST, porcine Sertoli cells. $^{**}P < 0.001$; ns, no significant. _P_ values were determined by two-sided Student's t-test.
(TIF)

**S5 Fig. The canonical autophagy pathway does not participate in inhibiting TGEV replication.** (Left) Western blot analysis of LC3-I and LC3-II in untreated WT and ATG5 KO cells; (Middle) Western blot analysis of LC3-I and LC3-II in WT and ATG5 KO cells after EBSS medium (EBSS) starvation and E64d supplementation for 6 h; (Right) Western blot analysis of TGEV N in WT and ATG5 KO cells following infection with TGEV (MOI = 1) at 12 hpi. GAPDH was shown as a loading control. WT, wild-type; KO, knockout; GAPDH, glyceraldehyde-3-phosphate dehydrogenase; TGEV, Transmissible gastroenteritis virus infected cells; Mock, uninfected cells; E64D, aloxistatin; ATG5, autophagy related gene 5.
(TIF)

**S6 Fig. Protein-protein interaction (PPI) network of the selected DEGs visualized with Cytoscape (WT-MOCK vs. WT-TGEV).** Differentially expressed genes in the "interferon signaling" pathway were marked with yellow. WT, wild-type; TGEV, Transmissible gastroenteritis virus infected cells; Mock, uninfected cells.
(TIF)

**S7 Fig. Protein-protein interaction (PPI) network of the selected DEGs visualized with Cytoscape (TMEM41B-KO-MOCK vs. TMEM41B-KO-TGEV).** Differentially expressed genes in the "interferon signaling" pathway were marked with yellow. KO, knockout; TGEV, Transmissible gastroenteritis virus infected cells; Mock, uninfected cells.
(TIF)

**S8 Fig. Protein-protein interaction (PPI) network of the selected DEGs visualized with Cytoscape (WT-MOCK vs. TMEM41B-KO-MOCK).** Differentially expressed genes in the "interferon signaling" pathway were marked with yellow. WT, wild-type; KO, knockout; Mock, uninfected cells.
(TIF)

**S9 Fig. Protein-protein interaction (PPI) network of the selected DEGs visualized with Cytoscape (WT-TGEV vs. TMEM41B-KO-TGEV).** Differentially expressed genes in the "interferon signaling" pathway were marked with yellow. WT, wild-type; KO, knockout; TGEV, Transmissible gastroenteritis virus infected cells.
(TIF)

**S10 Fig. Evaluation of the ability of TMEM41B to regulate the expression of ISGs in different types of cells.** (**A** and **C**) DNA sequence analysis showed that the presence of the mutation in (**A**) HEK293T and (**C**) L929 single clone originated TMEM41B KO cells. (**B** and **D**) Detection of mRNA expression of the ISGs in (**B**) TMEM41B-KO HEK293T and (**D**) TMEM41B-KO L929 cells by RT-qPCR. sgRNA, small guide RNA; PAM, protospacer adjacent motif; bp, base pairs; ins, insertion; del, deletion; WT, wild-type; KO, knockout. $^{*}P < 0.05$; $^{**}P < 0.01$; $^{***}P < 0.001$. $P$ values were determined by two-sided Student's t-test. Data are representative of at least three independent experiments.
(TIF)

**S11 Fig. TMEM41B is a negative regulator of cell-intrinsic immunity.** (**A-E**) mRNA levels of the *ISG*s (**A**) *MX1*, (**B**) *OASL*, (**C**) *IFIT1*, (**D**) *IFI6* and (**E**) *RASD2* on PK-15 cells or PK-15 cells overexpressing *TMEM41B* with or without PolyI:C stimulation. (**F-H**) mRNA levels of *ISG*s (**F**) *IFIT1*, (**G**) *RASD2* and (**H**) *OASL* in TMEM41B and IFNAR double KO cells were reduced to WT levels compared with TMEM41B KO cells. Data are representative of at least three independent experiments. NTC, negative control; PolyI:C, Polyinosinic: polycytidylic acid; WT, wild-type; KO, knockout. $^{*}P < 0.05$; $^{**}P < 0.01$; $^{***}P < 0.001$; ns, no significant. $P$ values were determined by two-sided Student's t-test. Data are representative of at least three

independent experiments.
(TIF)

**S12 Fig. Knockout TMEM41B inhibits the formation of DMVs induced by NSP3 and NSP4 from SARS-CoV-2.** (**A**) Schematic illustration of the backbone of SARS-CoV-2 Nsps plasmids. First, we generated TMEM41B KO HEK293T cell lines. Subsequently, we constructed PLV-mCherry-SARS-CoV-2-NSP3 and PLV-eGFP-SARS-CoV-2-NSP4 lentiviral expression vectors. After co-infection of PLV-mCherry-SARS-CoV-2-NSP3 and PLV-eGFP-SARS-CoV-2-NSP4 lentivirals in TMEM41B KO and WT HEK293T cells, the double immunofluorescence cells (mCherry and eGFP) were sorted by flow cytometer. (**B-F**) (**B**) HEK293T MOCK-treated cells and (**C-F**) SARS-CoV-2 NSP3 and NSP4 co-expressed cells were fixed for TEM analysis. (**C**) DMVs and (**D-F**) large deformation of ER can be observed. Red arrows indicate curved ERs. (**G-H**) HEK293T-TMEM41B-KO MOCK treatment cells (**G** and **H**) and SARS-CoV-2 NSP3 and NSP4 co-expressed cells (I-L) were fixed for TEM analysis. Mock, untreated control cells; +SARS2-CoV2-NSP3, NSP4, co-overexpression of NSP3 and NSP4 from SARS-CoV-2; WT, wild-type; KO, knockout; M, Mitochondria; ER, Endoplasmic reticulum; N, Nucleus; LD, lipid droplet; A, autophagosome; LTR, long terminal repeat; CMV, cytomegalovirus promoter; WPRE, Woodchuck Hepatitis Virus (WHP) Posttranscriptional Regulatory Element; GFP; green fluorescent protein; mCherry, mCherry fluorescent protein. Mock, uninfected cells.
(TIF)

**S13 Fig. The generation and characterization of TMEM41B-deficient mice.** (**A**) A schematic diagram of generation of F2 TMEM41B-deficient mice. (**B** and **C**) There was no significant morphological difference between F2 generation TMEM41B$^{+/-}$ mice and WT (C57BL/6N) mice. (**D**) Genotyping of F2 generation TMEM41B$^{+/-}$ mice using genomic DNA from tail clips. (**E**) Sanger sequencing of PCR products from F2 generation TMEM41B$^{+/-}$mice. (**F**) The relative *TMEM41B* gene expression level of F2 generation TMEM41B$^{+/-}$ mice liver tissue was reduced by about 70% compared with WT mice liver tissue. (**G**) Western blot assays analysis the autophagy maker p62 protein of liver and kidney tissue from WT and TMEM41B$^{+/-}$ mice with starvation treatment. (**H**) The serum lipid concentrations of WT and TMEM41B$^{+/-}$ mice (before MHV infection) were measured by corresponding reagent kit: TG (Rayto, S03027); CHO (Rayto, S03042); HDL (Rayto, S03025); LDL (Rayto, S03029). F, forward primer; R, reverse primer; WT, wild-type; NC, negative control; del, deletion, E1-7, exons from 1 to 7; bp, base pairs; M, DNA marker; sgRNA, small guide RNA; F0, founder generation mice; F1, F1 generation mice; F2, F2 generation mice; TG, triglyceride; CHO, cholesterol; HDL, High-density lipoprotein; LDL, Low-density lipoprotein. $^{*}P < 0.05$; $^{**}P < 0.01$; $^{***}P < 0.001$; ns, no significant. *P* values were determined by two-sided Student's t-test. Data are representative of at least three independent experiments.
(TIF)

**S1 Data. Sequencing results of sgRNAs (PigGeCKO) in each of three rounds of TGEV screening after challenge.**
(XLSX)

**S2 Data. Sequencing results of sgRNAs (Focused-CRISPR library) in the third and fifth rounds of TGEV screening after challenge.**
(XLS)

**S3 Data. Differential gene expression analysis derived from RNA-seq.**
(XLSX)

**S4 Data. Primer pairs and sgRNA targeting sequences used in this study.**
(XLSX)

## Acknowledgments

We thank the following investigators for contributing viral stocks: Prof. Hongbo Zhou (IAV PR8), Prof. Min Cui (JEV P3), Prof. Gang Cao (VSV and PRV), Prof. Shaobo Xiao (PDCoV: KT366560). We thank Dr. Yan Wang (Institute of Hydrobiology, Chinese Academy of Sciences) for her assistance with flow cytometry.

## Author Contributions

**Conceptualization:** Limeng Sun, Shuhong Zhao, Shengsong Xie, Guiqing Peng.

**Data curation:** Changzhi Zhao, Jingjin Li, Xiongwei Nie, Shengsong Xie.

**Formal analysis:** Changzhi Zhao, Zhen Fu, Yanan Fu, Zhelin Su, Yuan Zhou, Yubei Tan, Jingjin Li, Yixin Xiang, Shengsong Xie.

**Funding acquisition:** Shengsong Xie, Guiqing Peng.

**Methodology:** Limeng Sun, Changzhi Zhao, Yangyang Li, Fei Liu.

**Project administration:** Guiqing Peng.

**Software:** Xiongwei Nie.

**Supervision:** Guiqing Peng.

**Validation:** Limeng Sun, Changzhi Zhao, Zhen Fu, Yanan Fu, Zhelin Su, Yuan Zhou, Yubei Tan, Yixin Xiang, Jinfu Zhang.

**Visualization:** Limeng Sun, Xiongwei Nie.

**Writing – original draft:** Limeng Sun, Shengsong Xie.

**Writing – review & editing:** Limeng Sun, Changzhi Zhao, Shengsong Xie.

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
