## [Decision Letter · Decision Letter 0]

21 Jun 2021

Dear Dr. Guiqing,

Thank you very much for submitting your manuscript " Genome-scale CRISPR screen identifies TMEM41B as a multi-function host factor required for coronavirus replication " for consideration at PLOS Pathogens. As with all papers reviewed by the journal, your manuscript was reviewed by members of the editorial board and by several independent reviewers. In light of the reviews (below this email), we would like to invite the resubmission of a significantly-revised version that takes into account the reviewers' comments.

As noted by the reviewers, there are a number of experimental and methodological details that are missing that must be addressed. Concerns about distinctions between virus internalization and replication must be clarified. The points raised by Reviewer 1 are particularly insightful, including addressing inconsistencies in TGEV replication in TMEM41B KO cells. The reviewers request additional experimentation that would be required for reconsideration.

We cannot make any decision about publication until we have seen the revised manuscript and your response to the reviewers' comments. Your revised manuscript is also likely to be sent to reviewers for further evaluation.

Sincerely,

Sabra L. Klein

Associate Editor

PLOS Pathogens

Mark Heise

Section Editor

PLOS Pathogens

Kasturi Haldar

Editor-in-Chief

PLOS Pathogens

orcid.org/0000-0001-5065-158X

Michael Malim

Editor-in-Chief

PLOS Pathogens

orcid.org/0000-0002-7699-2064

As noted by the reviewers, there are a number of experimental and methodological details that are missing that must be addressed. Concerns about distinctions between virus internalization and replication must be clarified. The points raised by Reviewer 1 are particularly insightful, including addressing inconsistencies in TGEV replication in TMEM41B KO cells. The reviewers request additional experimentation that would be required for reconsideration.

Reviewer's Responses to Questions

**Part I - Summary**

Reviewer #1: Multiple independent studies using genome-wide CRISPR-Cas9 screens have identified TMEM41B as playing critical roles in autophagy and lipid mobilization. Subsequent screens to identify host factors required for viral replication identified TMEM41B as important for the replication of diverse coronaviruses and flaviviruses.

The authors extend the requirement of TMEM41B to a member of the Alphacoronavius family, TGEV, in a different host organism, Sus scrofa (Pig). Their findings are mostly in line with those previously reported in Hoffmann et al 2021. Novel findings from this study include the in vivo work and a potentially interesting association between TMEM41B and innate immune regulation. This manuscript is the first to demonstrate the functional relevance of TMEM41B and coronavirus disease pathogenesis in vivo. The authors propose a novel function for TMEM41B in regulating the innate immune response, however, additional evidence is needed to substantiate this function of TMEM41B.

Reviewer #2: This thorough submission presents results indicating that TMEM41B renders cells and experimental mice permissive to coronavirus infection. The study follows a standard path, from CRISPR KO screening, to identification and validation of TMEM41B as necessary for robust growth of virus, to focus toward mechanism of TMEM41B operations in virus infection cycles.

If there is a central issue, it is that very similar results have been published. The most strikingly similar work was published while this work was submitted, it is and https://doi.org/10.1371/journal.ppat.1009599 and so it is in the PPATH now. However, there may be good arguments for having some duplicated work in the journal.

The new findings in this submission are that (1) the TMEM41B suppresses interferon signaling, but this is not the proviral TMEM41B mechanism; (2) the TMEM41B is claimed to facilitate virus endocytosis; (3) the TMEM41B increases C57BL/6 mouse susceptibility to lethal MHV A59 infection. The main findings that TMEM41B is needed for post-entry replicative processes and that TMEM41B facilitates virus replication organelles are also present. These main findings concord with the previous publications on TMEM41B in flavivirus and coronavirus replication.

Reviewer #3: This manuscript reports the identification of TMEM41B as a host factor required for the infection of the transmissible gastroenteritis virus (TGEV), a member of α-coronaviruses. The authors show that TMEM41B is important for the formation of double-membrane vesicles that are derived from the ER and required for viral replication. The authors suggest that TMEM41B also protects mice from coronaviral (MHV) infection.

In general, it is important to search for host factors required for viral infections. However, at least six genome-wide CRISPR screening studies (one published in PLOS pathogens) have so far identified TMEM41B as a host factor of coronaviral infection, which reduces the novelty of this study. This study also provides potentially novel findings: (i) the effect of TMEM41B loss on the formation of double-membrane vesicles and (ii) the role of TMEM41B in mice in vivo. Unfortunately, the data of these two findings are not convincing enough in the present version (see specific comments below).

**Part II – Major Issues: Key Experiments Required for Acceptance**

Reviewer #1: Major points

If TMEM41B is the primary cause of the TGEV replication defect observed in the TMEM41B KO cells, why is there still such a drastic difference in TGEV replication when TMEM41B expression is restored in KO cells? The authors should include an explanation as to why TGEV replication is not rescued in the TMEM41B reconstituted cells. See Figure 2g-j: The relative levels of TMEM41B expression between TMEM41B-KO-rescue (figure 2g) and PK-15-overexpression (figure 2j) look comparable (relative to WT-NTC and PK-15-NTC, respectively) yet the TGEV mRNA detected in the RT-qPCR looks to differ by around 4 logs (figure 2h and j). TMEM41B-KO-rescue cells had 3 logs less TGEV mRNA detected relative to the WT-NTC (figure 2h)

The above point highlights the lack of information in the Methods section. For example, how did the authors perform the reconstitution experiments? From the Methods it seems that the KO clone constitutively expresses an sgRNA targeting TMEM41B. Did the authors mutate the TMEM41B cDNA so that it is resistant to cleavage? If the TMEM41B cDNA was reconstituted in cells expressing the TMEM41B sgRNA it could explain the partial rescue observed in the TMEM41B-KO-rescue cells.

The nomenclature in the manuscript is confusing. The authors use WT and PK-15 and it seems these may be the same but it is unclear. For example, what is the difference between figure 2g and 2i? In supplementary figure 10, overexpression of TMEM41B in PK-15 cells potently inhibits ISG expression compared to PK-15/WT cells, even after treatment with polyIC. Therefore, lower levels of the ISG transcripts for WT compared to the TMEM41B KO-rescue cells for figure 3f is unexpected. For example, we can see this in supplementary figure 10d – IFIT1 expression is lower in the PK-15-overexpression cells compared to WT in the absence of polyIC treatment. This should be comparable to IFIT1 in figure 3f, however, higher IFIT1 expression is observed in the TMEM41B-KO-rescue relative to WT. It is impossible to see if this is the case for the rest of the ISGs in figure 10 given the scale of the y-axis.

Figure 6b-f: The striking ISG upregulation observed in the TMEM41B KO cells confounds the replication defect observed for these virus families. The antiviral state in TMEM41B KO cells, beyond the function of TMEM41B, makes it difficult to conclude that the defect is due to the requirement of TMEM41B and not a result of the higher basal ISG levels. The authors address this for TGEV by generating a double KO for TMEM41B and IFNAR, which normalized the ISG expression, and confirm that TGEV replication is still not supported. Therefore, the effects seen for these other viruses could be due to IFN/ISG expression. This is particularly important for understanding the replication defect seen in negative strand-viruses which was not observed in a previous study. It is also mechanistically unclear how negative-strand RNA virus replication would be affected given the proposed role of TMEM41B in replication organelle formation for positive-strand viruses. The inconsistency between their model and how the negative-strand RNA viruses are inhibited is not explained and experiments to address this are not included. These results need to be confirmed in the double KO IFNAR cell lines as done with TGEV. Reconstitution experiments with TMEM41B with these other viral families should also be performed to demonstrate TMEM41B’s role in their replication.

Overall, the manuscript is difficult to follow, makes some inaccurate claims, and overstates results. Significant rephrasing of many statements is needed.

A few examples of many:

It is unclear what is meant by the statement made in lines 265-266:

“Additionally, we observed the subcellular localization of restore expression of TMEM41B in its KO cells infect with or without TGEV.”

Inaccurate statement on lines 247-248:

“CoVs, like all positive-strand RNA viruses of eukaryotes, hijack intracellular ER membranes to form their RO.”

Not all positive-strand RNA viruses replicate on ER membranes.

Although the observation that TMEM41B is required for TGEV replication in pig cells has some novelty (i.e. first report of TMEM41B requirement for this particular virus and host species), previous studies have demonstrated the importance of TMEM41B for the replication of various flaviviruses and coronaviruses in human, bovine, and insect cell lines. A comparison of the results highlighting the similarities and differences between the previous reports and the authors findings is largely absent from the paper. Also, most of the unique findings presented in this paper, particularly the ISG upregulation, are significantly underdeveloped.

Reviewer #2: The work is generally solid but there are some comments on the data:

1. Fig. 4d, these data do not provide much confidence in the claim that TMEM41B facilitates virus endocytosis. The figure lacks images of virus particles bound at 4C but not yet internalized by shift to 37C. Given equivalent virus binding to TMEM41B+ and – cells (Fig 4a), one wonders where the viruses are located in TMEM41B-KO cells in Fig 4d.

2. Fig. 5A, it is hard to be sure that the red arrow is pointing to a replication organelle; also, more replication organelles should be imaged to get a better sense of the TMEM41B role in RO formation. Optimally, immunogold EM or similar methods should be used to identify TMEM41B in the ROs.

3. Fig. 5bcd, the TMEM41B and TGEV NSP3 proteins are so over-expressed that it is hard to make claims for relevant colocalization. The IFA signals cover most of the cytoplasm. A suggestion is to assess colocalization at lower expression levels (levels consistent with natural expression in infected cells).

Reviewer #3: 1. The data of knockout mice (Fig. 7) are potentially important but not convincing.

- It is not clear how many mice were used in the experiments in a, b, and f, and what statistical analysis was used in a and b (the t-test is not acceptable). Actual p-values should be shown.

- It is questionable that heterozygous mice indeed have a defect in the function of TMEM41B. Even though heterozygous mice express a reduced amount of TMEM41B, it may be sufficient physiologically. Thus, it is essential to determine whether some of the key functions (e.g., p62 accumulation, autophagic flux, lipid droplet accumulation, coronaviral replication, DMV formation) are indeed defective in cultured TMEM41B+/- cells. Also, the p62 level, lipid droplet accumulation, and serum lipid concentrations should be compared between Wild-type and TMEM41B+/- mice (before infection).

- The authors used 6 mice in the experiment in Panel C. How can there be a 20% reduction?

- The authors should indicate the generation (F0, F1, or F2 etc.) of the mice used in Fig. 7 and Sup. Fig. 11C-F.

- The description "TMEM41B-KO" is inappropriate. This should be "TMEM41B+/-".

2. The quality of the electron microscopy data is not good. The images are too small to evaluate. In Fig. 5a, "double membranes" are not visible. Images with higher resolution should be presented. In addition, some quantification is required in order to show that TMEM41B is required for the formation of double-membrane vesicles.

3. It is not clear how the authors distinguish between viral internalization and replication. They use essentially the same method at different time points. This reviewer is not sure whether this is sufficient.

**Part III – Minor Issues: Editorial and Data Presentation Modifications**

Reviewer #1: Lines 119-120: “Our screen found that a total of 335 unique sgRNA sequences, which targeting to 317 unique protein-coding genes”

There seems to be no convergence of sgRNAs per gene identified in the screen – essentially one guide per gene. One would expect that with multiple sgRNAs targeting a single gene, there should be multiple sgRNAs enriched for each of the hits. This weakens the reliability of the screen data. Was the screen performed with replicates? At what fold-representation was the screen performed? There is no information about the full-genome screen in the Methods section.

Based on the results presented, TMEM41B was not the most reliable hit from the screens. What was the rationale for choosing TMEM41B as opposed to the other genes identified as top hits in the screen?

Figure 2f-j: What time point were the cells harvested at?

Figure 2g-j: What is the difference between WT-NTC and PK-15-NTC? If these are the same consistent naming would helpful.

Figure 3e and f: why are there such large discrepancies between the relative fold change between figures e and f? For example, there is 100-fold difference for MX1 when comparing WT and TMEM41B KO in figure 3e but only a 10-fold difference for MX1 in figure 3f. Would be nice to have an idea of the raw transcript numbers for these ISGs and see how this may influence the large fold changes observed in these experiments.

Figure 3: are there differences in IFN mRNA levels between the various cell lines? Does this change if TMEM41B is overexpressed?

Figure 3: Other cell lines should be tested to see the ISG upregulation observed in the TMEM41B KO cells is cell line specific. This would help to explain why their results differ from those previously reported.

Supplementary figure 10: It is surprising that the PK-15 cells overexpressing TMEM41B completely inhibits the upregulation of ISGs when treated with polyIC. Further work to show if this is polyIC specific or to help explain these results should be demonstrated.

Figure 4c: The low image resolution/magnification makes it impossible to state that no DMV formation is observed in the TMEM41B KO cells. We can see the absence of viral particles in the TMEM41B KO cells compared to the WT, but it cannot be determined from this image if DMVs are present.

Figure 4: The authors suggest viral endocytosis may contribute to the replication defect observed. It has been shown in other studies that when viral RNA is directly transfected into cells (bypassing endocytosis) a replication defect is still observed in TMEM41B KO cells. This highlights the need for Sun et al to compare their results to previous reports that used different viruses and host species and discuss the similarities and differences.

Reviewer #2: There are also some minor comments:

1. Fig. 1; consider highlighting the TMEM41B in panel b and d scatter plots

2. Fig. 4b, hard to be sure that the red arrows are pointing to virus particles (but evidence for less virus morphogenesis in TMEM41B-KO is strong from other results)

There are also some comments on the text:

1. The text has many confusing sentences, too many to itemize. Some examples include lines 94-100; this section summarizes results but does so in an awkward way. Authors would benefit from rephrasing this part. Lines 160-161; VTT domain is not in TGEV, it is in the TMEM41B related proteins. Rephrase this. Lines 200-202, requires rephrasing. Lines 221-222, this sentences is actually misleading and does not summarize the actual result, rephrase to emphasize that ratios of extra vs intra-cellular viral PFU are being assessed. Lines 374-375; requires rephrasing. There are more issues and there should be another go at revising the text.

2. The discussion section does not make any speculations on the roles that TMEM41B might have during minus strand RNA virus and DNA virus replication. Instead, there are only the statements on lines 388-390. There is an unrealized opportunity here.

Reviewer #3: 1. The recently identified role of TMEM41B as a lipid scramblase should be included in Discussion (doi: 10.1073/pnas.2101562118, doi: 10.1083/jcb.202103105, doi: 10.1016/j.cmet.2021.05.006).

2. Sequence IDs and alignment methods should be described in the legends or Methods.

3. Supplementary Fig. 5: Small but significant amounts of LC3-II are observed in ATG5-KO and ATG7-KO cells. These are not typical results, raising a concern that isolation of KO cells was not appropriate. Please carefully check them. Otherwise, the authors' claim that "the canonical autophagy pathway is not required for TGEV replication" (L184-185) is not valid.

PLOS authors have the option to publish the peer review history of their article (what does this mean?). If published, this will include your full peer review and any attached files.

Reviewer #1: No

Reviewer #2: No

Reviewer #3: No
---

## [Decision Letter · Decision Letter 1]

24 Sep 2021

Dear Dr. Guiqing,

Thank you very much for submitting your manuscript "Genome-scale CRISPR screen identifies TMEM41B as a multi-function host factor required for coronavirus replication" for consideration at PLOS Pathogens. As with all papers reviewed by the journal, your manuscript was reviewed by members of the editorial board and by several independent reviewers. The reviewers appreciated the attention to an important topic. Based on the reviews, we are likely to accept this manuscript for publication, providing that you modify the manuscript according to the review recommendations.

Requests for data to be shown in the actual manuscript or supplementary files, instead of just mentioned in text or only shown in the response to the review are justified and should be fixed.

Sincerely,

Sabra L. Klein

Associate Editor

PLOS Pathogens

Mark Heise

Section Editor

PLOS Pathogens

Kasturi Haldar

Editor-in-Chief

PLOS Pathogens

orcid.org/0000-0001-5065-158X

Michael Malim

Editor-in-Chief

PLOS Pathogens

orcid.org/0000-0002-7699-2064

Requests for data to be shown in the actual manuscript or supplementary files, instead of just mentioned in text or only shown in the response to the review are justified and should be fixed.

Reviewer Comments (if any, and for reference):

Reviewer's Responses to Questions

**Part I - Summary**

Reviewer #2: The authors responded strongly to previous reviews by providing additional results that significantly reinforce the key conclusions of the paper. They complemented TMEM41B KO cells with TMEM41B genes, did this properly, and this added important results demonstrating TMEM41B as a central virus permissively factor. They added new results that now show virus endocytosis and virus-induced DMVs at higher resolution, thereby supporting the contention that TMEM41B operates to support entry and DMV formation. They nicely illustrated distinct functions for TMEM41B in innate immune response, virus endocytosis, virus replication organelle formation, and in vivo virus infection in the liver. They included procedural details that now make it possible for readers to fully appreciate the experiments and results. They appropriately extended discussion of the findings and their contributions to the field.

Reviewer #3: The authors addressed most of my previous comments, but there remain some issues to be addressed.

Reviewer #4: The authors address many of the reviewer concerns in the revised manuscript. As before, the strengths of this manuscript include the in vivo work and the novel innate immune phenotype observed in the TMEM41B knockout (KO) PK-15 cells. The additional results with TMEM41B/IFNAR double KO PK-15 cells supports the role of TMEM41B in suppressing innate immune activation in PK-15 cells; however, as the authors point out, the effect is minor in other cell types. Indeed, the results in HEK293T and L929 cells are less convincing without additional controls. The ISG upregulation in TMEM41B KO cells may therefore not be generalizable to other cell types. Further, there is a missed opportunity to test other viruses in the TMEM41B/IFNAR1 double KO cells to rule out the possibility that reduced virus replication in TMEM41B KO cells results from elevated ISGs. Nevertheless, the innate immune signature in TMEM41B KO PK-15 cells will be of interest to the field and is worthy of further investigation. In the Discussion section, the authors may wish to speculate more on possible mechanisms by which TMEM41B regulates innate immunity.

Comments on revised manuscript

Major point 1:

The authors addressed the TMEM41B-KO-resuce by generating a stable single-clone cell line reconstituted with WT TMEM41B. They also include more detail on the methods used to generate the cell line.

TMEM41B is an intrinsic regulator of innate immunity

The authors also test human HEK293T and mouse L929 cell lines to determine if the innate immune regulation is observed in other cell types. These data are questionable without additional controls. The gene expression differences are minor and can easily be explained by clone-to-clone variability. As a result, it remains unclear if TMEM41B’s role in suppressing innate immunity is specific to PK-15 cells. Nevertheless, experiments conducted in this manuscript support the author’s claim of innate immune regulation by TMEM41B in PK-15 cells.

Major point 3:

The authors addressed the concern for figure 6B-G by testing the viruses in TMEM41B reconstituted cells and testing VSV in the TMEM41B/IFNAR double KO cell line. It would be nice to have conducted the TMEM41B reconstitution experiment in the TMEM41B/IFNAR double KO cell line since WT is not the best comparison after generating and growing out single cell clones (particularly after multiple rounds as would have been done for the double KO cell line). Again, there seems to be some missed opportunity here as the innate immune phenotype is one of the more interesting findings in the study. It leaves the reader to wonder how IAV infects the double KO cell line.

**Part II – Major Issues: Key Experiments Required for Acceptance**

Reviewer #2: The opinion of this reviewer is that major issues have now been resolved and key experiments required for acceptance have now been completed.

Reviewer #3: Previous Major Point 1.

- The authors now show that the levels of serum lipids and tissue LC3-II are reduced (Fig. S13). Although the quantitative data of serum TG and LDL cholesterol levels are convincing, the authors’ claim that autophagic activity is reduced is not supported by these data. It has been shown that LC3-II rather accumulates in TMEM41B-/- cells (Ref #34-36). Therefore, the reduction of LC3-II in TMEM41B+/- tissues does not indicate autophagy suppression. Rather, it is actually the opposite. The authors have not included any other autophagic flux data (e.g., p62, LC3 turnover, GFP-RFP-LC3, etc.). Without such data, it would be fair to state that “while serum lipid levels are reduced, autophagic activity may not be affected in heterozygous mice”.

- The information of the number of mice and the method of statistics is still lacking in the legend to Fig. 7A, B and F. Please include.

Previous Major Point 2.

- The results of the quantification of DMVs should be shown in the manuscript, not only in the rebuttal letter.

Reviewer #4: (No Response)

**Part III – Minor Issues: Editorial and Data Presentation Modifications**

Reviewer #2: 1. The authors have done a good job at determining whether TMEM41B confers cell susceptibility (virus entry, Figs 4AF) or cell permissively (virus replication, Fig 4GH and others). While they have not fully quantified the relative TMEM41B support for each step, the data point strongly to the replication stage being the one that is most dependent on TMEM41B, and likely the most biologically relevant stage. This could be emphasized in the discussion section.

2. Lines 318-327 are discussion points and belong there, not in the results section.

3. The text will need editing for syntax corrections.

4. The figures can be clarified by matching color codes between panels, i.e., in Fig 7 and several other figs.

Reviewer #3: (No Response)

Reviewer #4: (No Response)

PLOS authors have the option to publish the peer review history of their article (what does this mean?). If published, this will include your full peer review and any attached files.

Reviewer #2: No

Reviewer #3: No

Reviewer #4: No

Figure Files:

Data Requirements:

Reproducibility:

References:

---

## [Editor Report · Decision Letter 2]

14 Nov 2021

Dear Dr. Guiqing,

We are pleased to inform you that your manuscript 'Genome-scale CRISPR screen identifies TMEM41B as a multi-function host factor required for coronavirus replication' has been provisionally accepted for publication in PLOS Pathogens.

Best regards,

Sabra L. Klein

Associate Editor

PLOS Pathogens

Mark Heise

Section Editor

PLOS Pathogens

Kasturi Haldar

Editor-in-Chief

PLOS Pathogens

orcid.org/0000-0001-5065-158X

Michael Malim

Editor-in-Chief

PLOS Pathogens

orcid.org/0000-0002-7699-2064
---

## [Editor Report · Acceptance letter]

29 Nov 2021

Dear Dr. Guiqing,

We are delighted to inform you that your manuscript, " Genome-scale CRISPR screen identifies TMEM41B as a multi-function host factor required for coronavirus replication ," has been formally accepted for publication in PLOS Pathogens.

Best regards,

Kasturi Haldar

Editor-in-Chief

PLOS Pathogens

orcid.org/0000-0001-5065-158X

Michael Malim

Editor-in-Chief

PLOS Pathogens

orcid.org/0000-0002-7699-2064